# Single-cell analysis of the ventricular-subventricular zone reveals signatures of dorsal and ventral adult neurogenesis

Arantxa Cebrian Silla[1,2†], Marcos Assis Nascimento[1,2†], Stephanie A Redmond[1,2†], Benjamin Mansky[1,2,3†], David Wu[1,2,4,5], Kirsten Obernier[1,2], Ricardo Romero Rodriguez[1,2], Susana Gonzalez Granero[6], Jose Manuel García-Verdugo[6], Daniel A Lim[1,2], Arturo Álvarez-Buylla[1,2*]

[1]Eli and Edythe Broad Institute for Stem Cell Research and Regeneration Medicine, University of California, San Francisco, San Francisco, United States; [2]Department of Neurological Surgery, University of California, San Francisco, San Francisco, United States; [3]Neuroscience Graduate Program, University of California, San Francisco, San Francisco, United States; [4]Medical Scientist Training Program, University of California, San Francisco, San Francisco, United States; [5]Biomedical Sciences Graduate Program, University of California, San Francisco, San Francisco, United States; [6]Laboratorio de Neurobiología Comparada, Instituto Cavanilles de Biodiversidad y Biología Evolutiva, Universitat de València - Centro de Investigación Biomédica en Red sobre Enfermedades Neurodegenerativas (CIBERNED), Paterna, Spain

*For correspondence:
abuylla@stemcell.ucsf.edu

†These authors contributed equally to this work

**Abstract** The ventricular-subventricular zone (V-SVZ), on the walls of the lateral ventricles, harbors the largest neurogenic niche in the adult mouse brain. Previous work has shown that neural stem/progenitor cells (NSPCs) in different locations within the V-SVZ produce different subtypes of new neurons for the olfactory bulb. The molecular signatures that underlie this regional heterogeneity remain largely unknown. Here, we present a single-cell RNA-sequencing dataset of the adult mouse V-SVZ revealing two populations of NSPCs that reside in largely non-overlapping domains in either the dorsal or ventral V-SVZ. These regional differences in gene expression were further validated using a single-nucleus RNA-sequencing reference dataset of regionally microdissected domains of the V-SVZ and by immunocytochemistry and RNAscope localization. We also identify two subpopulations of young neurons that have gene expression profiles consistent with a dorsal or ventral origin. Interestingly, a subset of genes are dynamically expressed, but maintained, in the ventral or dorsal lineages. The study provides novel markers and territories to understand the region-specific regulation of adult neurogenesis.

## Introduction

Neural stem/progenitor cells (NSPC) persist in the adult mouse brain in the walls of the forebrain ventricles. This neurogenic niche includes the ventricular-subventricular zones in the walls of the lateral ventricles (V-SVZ), home to a subpopulation of astrocytes (B cells) that function as the NSPCs (*Chaker et al., 2016*; *Doetsch et al., 1997*; *Ihrie and Alvarez-Buylla, 2011*; *Lim and Alvarez-Buylla, 2014*; *Mirzadeh et al., 2008*). This neurogenic region has also been referred to as the SVZ or the subependymal zone (*Kazanis et al., 2017*). B cells generate intermediate progenitors (C cells) that, in turn, give rise to neuroblasts (A cells) that migrate to the olfactory bulb (OB) (*Obernier et al., 2018*; *Ponti et al., 2013*). A subpopulation of B cells also generate

**eLife digest** Nerve cells, or neurons, are the central building blocks of brain circuits. Their damage, death or loss of function leads to cognitive decline. Neural stem/progenitor cells (NSPCs) first appear during embryo development, generating most of the neurons found in the nervous system. However, the adult brain retains a small subpopulation of NSPCs, which in some species are an important source of new neurons throughout life.

In the adult mouse brain the largest population of NSPCs, known as B cells, is found in an area called the ventricular-subventricular zone (V-SVZ). These V-SVZ B cells have properties of specialized support cells known as astrocytes, but they can also divide and generate intermediate 'progenitor cells' called C cells. These, in turn, divide to generate large numbers of young 'A cells' neurons that undertake a long and complex migration from V-SVZ to the olfactory bulb, the first relay in the central nervous system for the processing of smells.

Depending on their location in the V-SVZ, B cells can generate different kinds of neurons, leading to at least ten subtypes of neurons. Why this is the case is still poorly understood.

To examine this question, Cebrián-Silla, Nascimento, Redmond, Mansky et al. determined which genes were expressed in B, C and A cells from different parts of the V-SVZ. While cells within each of these populations had different expression patterns, those that originated in the same V-SVZ locations shared a set of genes, many of which associated with regional specification in the developing brain. Some, however, were intriguingly linked to hormonal regulation.

Salient differences between B cells depended on whether the cells originated closer to the top ('dorsal' position) or to the bottom of the brain ('ventral' position). This information was used to stain slices of mouse brains for the RNA and proteins produced by these genes in different regions. These experiments revealed dorsal and ventral territories containing B cells with distinct 'gene expression'.

This study highlights the heterogeneity of NSPCs, revealing key molecular differences among B cells in dorsal and ventral areas of the V-SVZ and reinforcing the concept that the location of NSPCs determines the types of neuron they generate. Furthermore, the birth of specific types of neurons from B cells that are so strictly localized highlights the importance of neuronal migration to ensure that young neurons with specific properties reach their appropriate destination in the olfactory bulb.

The work by Cebrián-Silla, Nascimento, Redmond, Mansky et al. has identified sets of genes that are differentially expressed in dorsal and ventral regions which may contribute to regional regulation. Furthering the understanding of how adult NSPCs differ according to their location will help determine how various neuron types emerge in the adult brain.

oligodendrocytes (*Figueres-Oñate et al., 2019*; *Gonzalez-Perez, 2014*; *Kazanis et al., 2017*; *Menn et al., 2006*; *Nait-Oumesmar et al., 1999*; *Picard-Riera et al., 2002*). From the initial interpretation that adult NSPCs are multipotent and able to generate a wide range of neural cell types (*Morshead et al., 1994*; *Reynolds and Weiss, 1992*; *van der Kooy and Weiss, 2000*), more recent work suggests that the adult NSPCs are heterogeneous and specialized, depending on their location, for the generation of specific types of neurons, and possibly glia (*Chaker et al., 2016*; *Delgado et al., 2020*; *Fiorelli et al., 2015*; *Merkle et al., 2014*, *Merkle et al., 2007*; *Tsai et al., 2012*). Previous single-cell sequencing experiments in the V-SVZ have described the many broad classes of cells that reside in the niche. For example, transcriptional analyses after cell sorting have identified stages in the B-C-A cell lineage (*Borrett et al., 2020*; *Codega et al., 2014*; *Dulken et al., 2017*; *Xie et al., 2020*), as well as populations of NSPCs that appear to activate after injury (*Llorens-Bobadilla et al., 2015*). Profiling of the entire niche has highlighted differences between quiescent and activated B cells (*Mizrak et al., 2020*; *Zywitza et al., 2018*). However, the differences among B cells of equivalent activation state (e.g. quiescent, primed, or activated) or the B cell heterogeneity that leads to the generation of diverse neuronal subtypes remain poorly understood.

NSPC heterogeneity, interestingly, is largely driven by their location within the adult V-SVZ. This concept explains why young neurons in the OB originate over such a wide territory in the walls of the lateral ventricles. Multiple studies have begun to identify regional differences in gene expression among the lateral, septal, and subcallosal walls of the lateral ventricles (*Delgado et al., 2021*). For

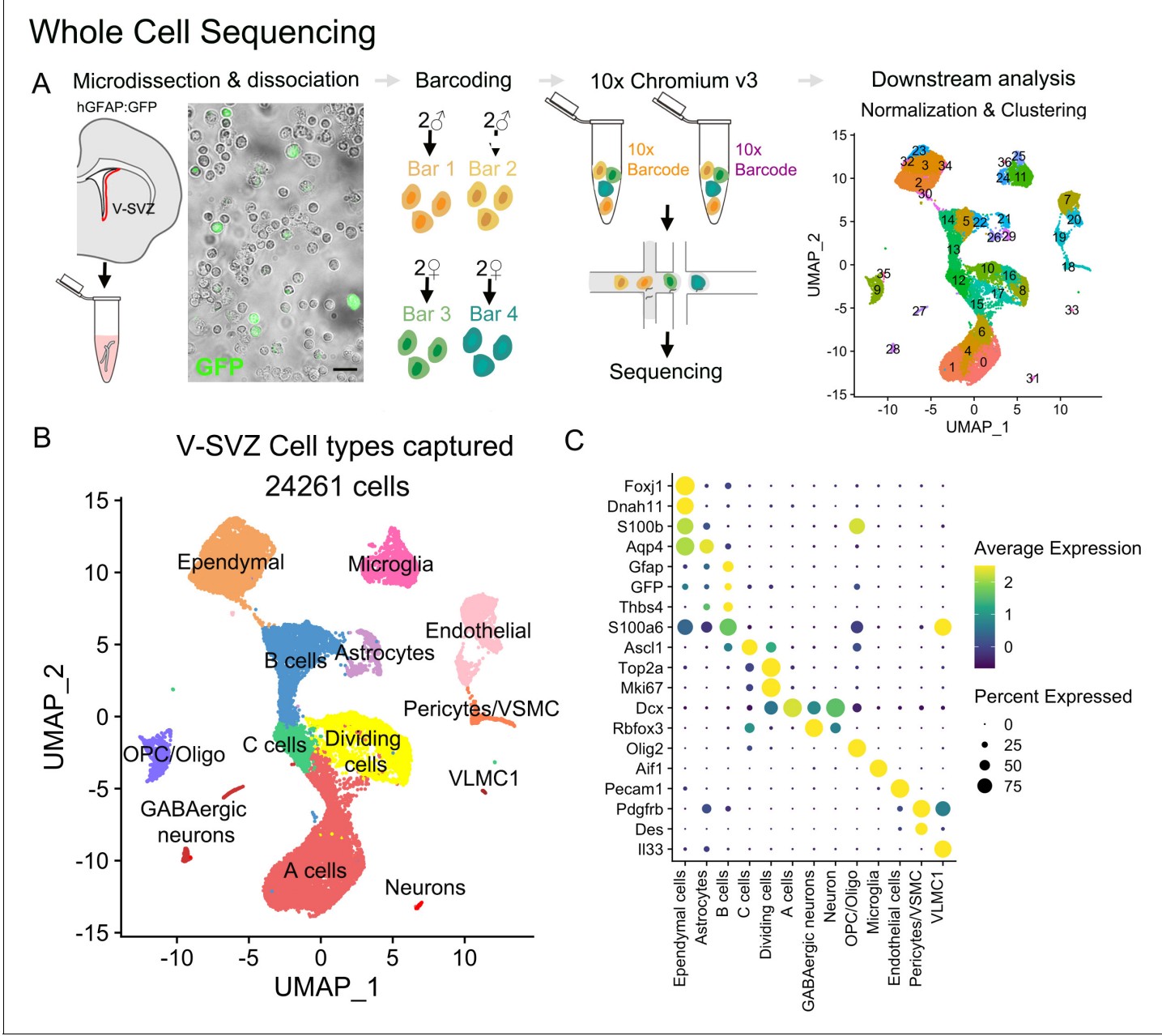

**Figure 1.** Whole-cell sequencing captures the cellular diversity and activation cascade of the adult neurogenic niche. (**A**) Schematic of the whole-cell single-cell isolation and sequencing protocol (scRNA-Seq). The lateral wall of the lateral ventricles were microdissected from young adult hGFAP:GFP mouse brains (n=four males, n=four females). Four samples were multiplexed with MULTI-seq barcodes and combined together. Two 10x Chromium Controller lanes were loaded as technical replicates, and cells were sequenced and processed for downstream analysis (*Figure 1—figure supplement 1*). V-SVZ: Ventricular-Subventricular zone. (**B**) UMAP plot of scRNA-Seq cell types captured after demultiplexing and doublet removal. (**C**) Dot plot of cell-type-specific marker expression in the clusters from (**B**) (*Figure 1—figure supplement 2*).

The online version of this article includes the following figure supplement(s) for figure 1:

**Figure supplement 1.** Biological and technical replicate metadata of the scRNA-Seq dataset.

**Figure supplement 2.** Transcriptomic profile of B cells versus Parenchymal Astrocytes.

example, differences in gene expression of B cells from the septal and lateral walls of the lateral ventricles have been recently observed (*Mizrak et al., 2019*). Other studies have shown that *Pax6* and *Hopx*-expressing cells correspond to dorsal V-SVZ progenitors (*Hack et al., 2005*; *Kohwi et al., 2005*; *Zweifel et al., 2018*), and *Vax1*-expressing young neuroblasts are derived from ventral

progenitors (*Coré et al., 2020*). Spatially defined lineage-tracing studies using microinjections of viruses have identified subdomains of the V-SVZ that give rise to specific subtypes of OB neurons (*Merkle et al., 2014*; *Merkle et al., 2007*; *Ventura and Goldman, 2007*). Lineage-tracing studies have demonstrated that these regional subdomains largely follow the territories defined by developmentally regulated transcription factors including Pax6, Nkx2.1, Nkx6.2, and Emx1 (*Delgado et al., 2020*; *Delgado and Lim, 2015*; *Kohwi et al., 2007*, *Kohwi et al., 2005*; *Merkle et al., 2014*; *Willaime-Morawek et al., 2006*; *Young et al., 2007*), but the molecular differences among B cells underlying their regionally-restricted potential are largely unknown.

Here, we have undertaken single-cell and single-nucleus RNA sequencing of the microdissected V-SVZ to gain insight into these important questions regarding NSPC heterogeneity and their developmental potential. Clustering analysis reveals strong dorso-ventral differences in lateral wall B cells. Validation of these differential gene expression patterns has revealed the anatomical boundary that separates these dorsal and ventral B cell domains. Additionally, our analysis identifies subpopulations of A cells defined by maturation state and dorso-ventral origin. We also find that a subset of dorso-ventral B cell transcriptional differences are retained through the C and A cell stages of the lineage. These new data advance our molecular understanding of how major region-specific neural lineages emerge in the adult V-SVZ and begin to delineate major functional subclasses of adult-born young neurons.

## Results

### Single-cell RNA sequencing distinguishes B cells from parenchymal astrocytes and reveals B cell heterogeneity

For whole single-cell RNA sequencing (scRNA-Seq), we dissected the lateral wall of the lateral ventricle from hGFAP:GFP mice at postnatal day (P) 29–35 (n=8, *Figure 1A*; *Figure 1—figure supplement 1A*). To determine possible sex differences in downstream analyses, two male and two female samples (n=four samples total) were dissociated and multiplexed by labeling cells with sample-specific MULTI-seq barcodes (*McGinnis et al., 2019*). Multiplexed samples were then pooled for the remainder of the single-cell isolation protocol. Two technical replicates of pooled samples were loaded in separate lanes of the Chromium Controller chip (10x Genomics) for single-cell barcoding and downstream mRNA library preparation and sequencing (*Figure 1A*). Cells carrying multiple barcodes or a high number of mRNA reads (4128 out of 35,025 cells, 11.7%) were considered doublets and were eliminated from downstream analysis. Data from each technical replicate were integrated for batch correction (*Stuart et al., 2019*). We then performed unbiased clustering of cell profiles and calculated UMAP coordinates for data visualization (*Figure 1A*). The clustering of lateral wall V-SVZ cells was not driven by sample, technical replicate, or sex (*Figure 1—figure supplement 1B–J*). Cell cluster identities were annotated based on the detection of known cell type markers (*Figure 1B–C*). We identified 37 clusters, with 14 clusters corresponding to cell types within the neurogenic lineage: NSPCs (B cells), intermediate progenitors (C cells), and neuroblasts (A cells) (*Doetsch et al., 1999*; *Obernier et al., 2018*). In addition, our analysis identified cell clusters corresponding to parenchymal astrocytes, ependymal cells, neurons, oligodendroglia, microglia, pericytes, vascular smooth muscle cells, and endothelial cells (*Figure 1B–C*).

NSPCs in the V-SVZ correspond to a subpopulation of astrocytes (B cells) derived from radial glia (*Doetsch et al., 1999*; *Laywell et al., 2000*; *Merkle et al., 2004*). B cells have ultrastructure and markers of astrocytes (*Borrett et al., 2020*; *Codega et al., 2014*). Therefore, identifying markers that distinguish parenchymal astrocytes from B cells has been a challenge in the field. A fraction of both populations expressed *Gfap*: 51.85% of B cells (clusters 5, 13, 14, and 22) and 24.37% of parenchymal astrocytes (clusters 21, 26, and 29). This is consistent with previous reports (*Chai et al., 2017*; *Ponti et al., 2013*; *Xie et al., 2020*). Note that across all cells captured in our scRNAseq analysis, only B cells, parenchymal astrocytes or ependymal cells expressed high levels of *Gfap*. Furthermore, among these three cell types, B cells had the highest average expression of *Gfap* (4.41 for B cells, 1.00 for astrocytes, 0.298 for ependymal cells, values in SCT corrected counts). Other markers, like *S100a6* (*Kjell et al., 2020*) (88.9% of B cells; 54% of parenchymal astrocytes, and 80% of ependymal cells) and *Thbs4* (*Zywitza et al., 2018*) (45% of B cells; 28.77% in parenchymal astrocytes, 2.88% in ependymal cells) are also expressed preferentially in B cells, but they alone do not

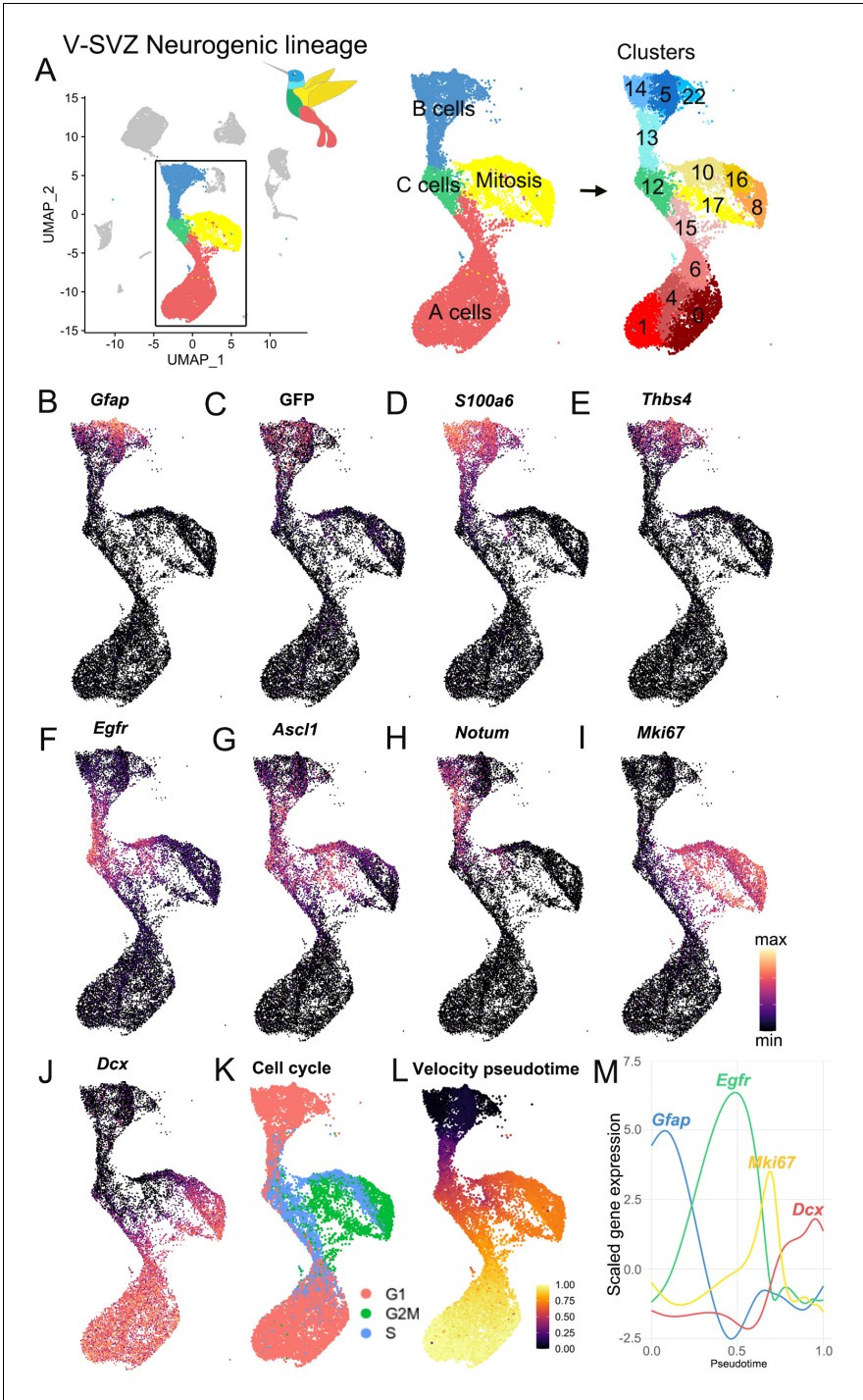

**Figure 2.** Characterization of the scRNA-Seq V-SVZ neurogenic lineage. (**A**) The neurogenic lineage has a 'bird-like' shape, with B cells forming the head (blue), C cells in the body (green), dividing cells in the wing (yellow), and A cells in the tail (red). These are divisible into 14 distinct clusters, including 4 B cell clusters, 1 C cell cluster, 4 clusters of dividing cells, and 5 A cell clusters (right). (**B-J**). Gene expression captures progression along the lineage, with canonical markers of each stage expressed in its corresponding region of the UMAP plot. (**K**) Scoring cells by phase of the cell cycle reveals cells in G2M and S phase occupying the wing of the bird. (**L-M**). Pseudotime calculated by RNA velocity recapitulates the B to C to A trajectory along the neurogenic lineage. (**M**) Genes associated with B cells (*Gfap*), activated B and C cells (*Egfr*), mitosis (*Mki67*), and A cells (*Dcx*) peak in expression at corresponding stages in the lineage.

distinguish these two cell populations (*Figure 1C*, *Figure 1—figure supplement 2A,B*). We performed differential gene expression analysis to further distinguish B cells from parenchymal astrocytes. We found that B cells had a higher expression of *Maff, Zfp36, Bex4, Lgals3,* and *Anxa2* compared to parenchymal astrocytes, which in turn were enriched for *Clmn, Atp13a4, Eps8, Pcdh7,* and *Syne1* (*Figure 1—figure supplement 2C*, *Supplementary file 1*).

To better understand the biological differences between parenchymal astrocytes and B cells, we performed gene ontology (GO) enrichment on the differentially expressed genes. Genes associated with synapse regulation (GO:0051965 and 0051963), macroautophagy (GO: 0016241), and dendrite development and morphogenesis (GO: 0016358 and 0048813), among others, were overrepresented in parenchymal astrocytes compared to B cells (*Figure 1—figure supplement 2D*). In contrast, B cells were enriched in terms associated with RNA regulation (GO:0000463, 0034471, 0000956, and 0000966) and mitochondrial regulation (GO:006626, 0073655, 0090201, and 0010823) (*Figure 1—figure supplement 2D*). These differentially represented GO terms support the known neuro-regulatory function of parenchymal astrocytes, as well as the increased transcriptional regulation that has been associated with the transition of NSPCs from quiescent to activated states (*Dulken et al., 2017*; *Llorens-Bobadilla et al., 2015*). Using this cell type classification, we included B cell clusters, but not striatal astrocytes, in our downstream analysis of the neurogenic lineage.

## scRNA-Seq captures neurogenic progression in the V-SVZ

In our scRNA-Seq dataset, we found that the majority of cells were part of the neurogenic lineage, which is composed of primary progenitors, intermediate progenitors, and young neurons (the B-C-A cell lineage). These neurogenic lineage clusters were in the center of the UMAP plot. With this analysis, the data had a 'hummingbird-like' shape with B cells (clusters 5, 13, 14, and 22) at the top in the bird's head and neck, proliferating cells (clusters 8, 10, 16, and 17) and C cells (cluster 12) resembled the bird's body and wing, and A cells (clusters 0, 1, 4, 6, and 15) formed a tail (*Figure 2A*). As expected, all B cell clusters expressed *Gfap, GFP,* and *S100a6* (*Figure 2B–D*). We identified a subpopulation of B cells as the quiescent B cells (*Codega et al., 2014*; *Llorens-Bobadilla et al., 2015*) (clusters 5, 14, and 22) characterized by high expression of *Thbs4* and *Gfap*, no *Egfr*, and low *Ascl1* expression (*Figure 2B,E,F,G*). In contrast, cluster 13 (the neck region of the 'hummingbird') corresponded to activated B cells with lower expression of *Gfap, GFP,* and *S100a6,* but high *Egfr* and *Ascl1* expression (*Codega et al., 2014*; *Figure 2B–D,F–G*). Cluster 13 cells also expressed *Notum*, a marker recently associated with activating qNSPCs (*Mizrak et al., 2020*; *Figure 2H*). Bordering cluster 13 in the chest region of the 'bird' was cluster 12, identified as the C cell cluster, which had low expression of astrocytic markers *Gfap* and *GFP* (*Figure 2B–C*), but high expression of *Egfr* and *Ascl1* (*Figure 2F–G*). The wing area of the 'bird' contained M*ki67*+ proliferating cells (*Figure 2I*), and in the tail area of the 'bird' below, clusters corresponding to neuroblasts were characterized by high expression of *Dcx*+ (*Figure 2J*). We used cell-cycle scoring to classify cells by their G2M, S, or G1 phase (*Tirosh et al., 2016*) and found that a high number of cells in cluster 12 were in the S phase (691/992, 69.6%, *Supplementary file 2*), consistent with cluster 12 corresponding to intermediate progenitors (C cells) (*Figure 2K*).

The unbiased clustering analysis pooled dividing cells into the wing region of the 'hummingbird'. A closer look at these clusters of mitotic cells in G2 or metaphase (G2M; clusters 10, 16, 17, and 8) showed that their gene expression pattern overlaps with that of the non-dividing neurogenic lineage cell progression: Cluster 10 expresses markers of B cells (*GFP, Notum*), cluster 16 had markers of C cells (*Ascl1*), and cluster 8 expresses markers of A cells (*Dcx*) (*Figure 2B–K*). This suggests that these clusters correspond to dividing B, C, and A cells, respectively, which have all been previously observed by electron and confocal microscopy in vivo (*Doetsch et al., 1997*).

Identified by *Dcx* expression, the largest number of cells within the neurogenic lineage corresponded to neuroblasts and young neurons (A cells) (clusters 0, 1, 4, 6, and 15; the tail region of the hummingbird) (*Figure 2A,J*). Interestingly, unbiased clustering subdivided A cells into five subclusters with different gene expression profiles. Consistent with previous work showing that a subpopulation of newly generated neurons continues to divide (*Lois and Alvarez-Buylla, 1993*; *Menezes et al., 1995*), clusters 6 and 15 contain *Dcx*+ A cells with proliferative markers (e.g. M*ki67*) (*Figure 2I–K*). Genes that distinguished clusters 0, 1, 4, 6, and 15 are discussed below. The overall progression from B-C-A cells described above is supported by RNA-velocity lineage trajectory

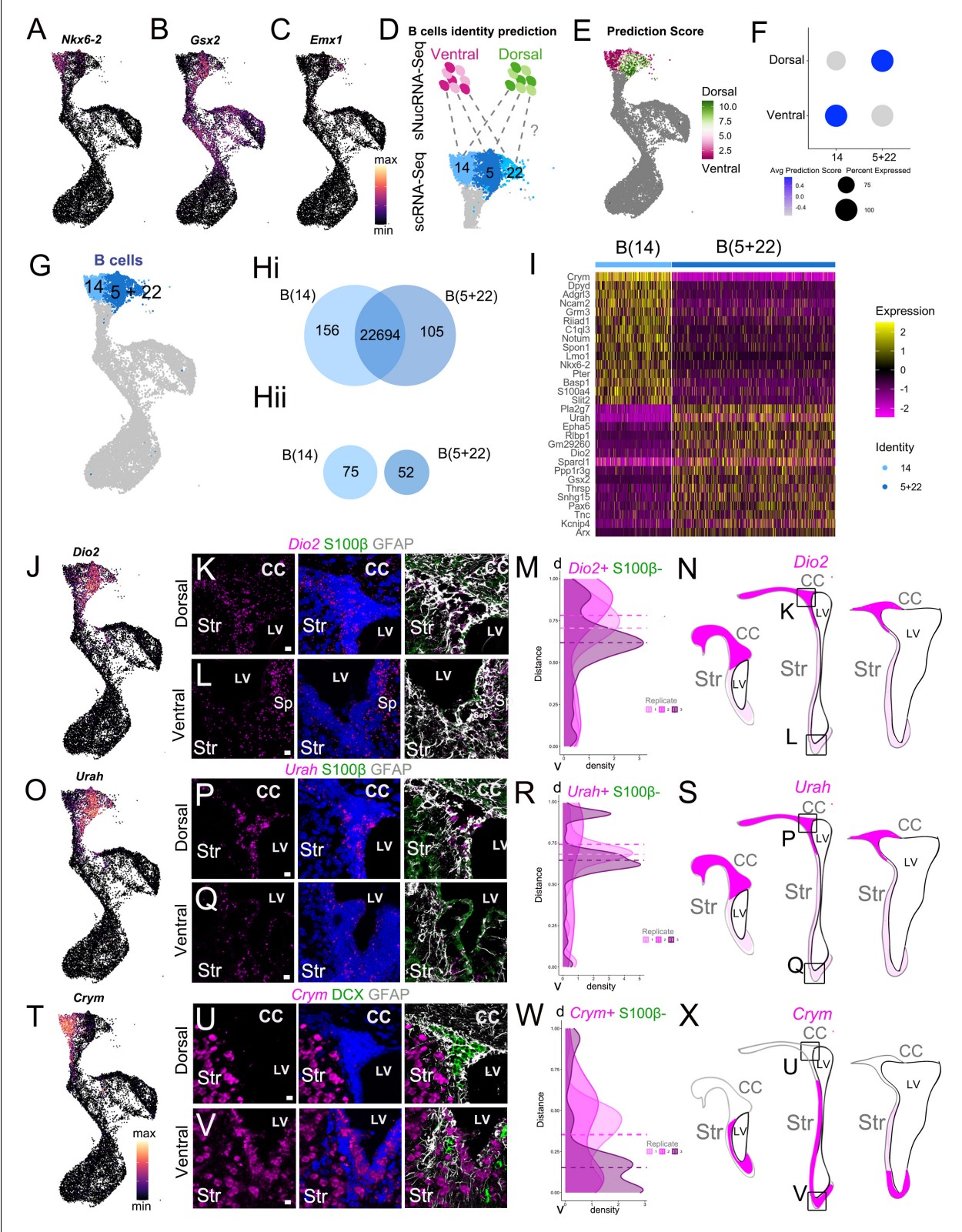

**Figure 3.** scRNAseq reveals regional heterogeneity among adult neural stem cells. (A-C) UMAP plots of *Nkx2.1* (A), *Gsx2* (B), and *Emx1* (C) expression in the scRNA-Seq neurogenic lineage (see also *Figure 3—figure supplement 1*). (D) Schematic of the region identity prediction calculation, where anchor gene sets (dashed gray lines) are calculated between Ventral Dissection (magentas) and Dorsal Dissection (greens) sNucRNA-Seq B cell nuclei and scRNAseq B cells (blues) and each scRNAseq B cell is given a Dorsal or Ventral Dissection predicted identity score (see also *Figure 3—figure*
*Figure 3 continued on next page*

*Figure 3 continued*

supplement 2). (**E**) The net predicted identity scores of each scRNA-Seq B cell plotted in UMAP space, where strongly Dorsal Dissection predicted identity cells are dark green, and strongly Ventral Dissection predicted identities are dark magenta. Dark gray cells were not included in the analysis. (**F**) Dot plot of the average Dorsal or Ventral Dissection predicted identity scores for scRNA-Seq B cell clusters B(14) and B(5+22). (**G**) UMAP plot of B cell cluster identities used in the following analysis: B(14) (light blue) and B(5+22) (dark blue). (**H**) (i) Venn diagram summarizing differential gene expression analysis between clusters B(14) (light blue) and B(5+22) (dark blue). (ii) Numbers of candidate marker genes identified after selecting significantly upregulated genes expressed in no more than 40% of cells of the other cluster. (**I**) Heatmap depicting expression of the top 10 differentially expressed genes between clusters B(14) (left) and B(5+22) (right). (**J**) UMAP plot of *Dio2* expression in the scRNA-Seq neurogenic lineage. (**K - L**) Confocal micrographs of *Dio2* RNA (magenta), S100b (green), and GFAP (white) protein expression in the dorsal (**K**) and ventral (**L**) V-SVZ. (**M-N**). Quantification of DAPI+, S100ß- *Dio2* RNAscope puncta along the length of the V-SVZ (0 = ventral-most extent, 1 = dorso-lateral-most extent of the wedge) with the median puncta distribution location plotted as a horizontal line for each sample (n=3, each indicated by a different shade, ~ bregma 1.18, 0.98 and 0.62 mm). N. Summary schematic of *Dio2* expression in three rostro-caudal coronal sections (~bregma 1.50, 0.98 and 0.14 mm); strong expression (magenta); sparse (light magenta). Boxed areas denote locations of dorsal (**K**) and ventral (**L**) high-magnification images. (**O**) UMAP plot of *Urah* expression in the scRNA-Seq neurogenic lineage. (**P - Q**) Confocal micrographs of *Urah* RNA (magenta), S100b (green), and GFAP (white) protein expression in the dorsal (**P**) and ventral (**Q**) V-SVZ. (**R-S**) Quantification of DAPI+, S100ß- *Urah* RNAscope puncta along the length of the V-SVZ (0 = ventral-most extent, 1 = dorso-lateral-most extent of the wedge) with the median puncta distribution location plotted as a horizontal line for each sample (n=3, each indicated by a different shade, bregma~ 1.34, 1.18 and 0.98 mm). S. Summary schematic of *Urah* expression in three rostro-caudal coronal sections (~bregma 1.50, 0.98 and 0.14 mm); strong expression (magenta); sparse (light magenta). Boxed areas denote locations of dorsal (**P**) and ventral (**Q**) high-magnification images. **T** UMAP plot of *Crym* expression in the scRNA-Seq neurogenic lineage. (**U-V**) Confocal micrographs of *Crym* RNA (magenta), DCX (green), and GFAP (white) protein expression in the dorsal (**U**) and ventral (**V**) V-SVZ. (**W-X**). Quantification of DAPI+,S100B- *Crym* RNAscope puncta along the length of the V-SVZ (as in M: 0 = ventral-most extent, 1 = dorso-lateral-most extent of the wedge; n=3, bregma~ 1.42, 1.18 and 0.98 mm). X. Summary schematic of *Crym* expression in three rostro-caudal coronal sections (~bregma 1.50, 0.98 and 0.14 mm); strong expression (magenta); low expression (white). Boxed areas denote locations of dorsal (**U**) and ventral (**V**) high-magnification images. DAPI: blue, LV: lateral ventricle, d: dorsal, v: ventral, CC: corpus callosum, Str: striatum. Scale bars: 10 µm (K, L, P, Q, U, and V).

The online version of this article includes the following source data and figure supplement(s) for figure 3:

**Source data 1.** Quantifications of *Crym*, *Dio2*, and *Urah* RNAscope spots.
**Figure supplement 1.** Identification of quiescent B cell clusters.
**Figure supplement 2.** Characterization of the sNucRNA-Seq dataset.
**Figure supplement 3.** Gene Regulatory networks of dorsal and ventral B cell markers.
**Figure supplement 3—source data 1.** Nodes and edges of gene regulatory networks of the top 10 dorsal or ventral DE genes in B cells.
**Figure supplement 4.** *Urah* and *Crym*-associated gene regulatory networks in dorsal and ventral B cells.
**Figure supplement 4—source data 1.** Nodes and edges of *Urah* or *Crym* gene regulatory networks in B cells.

reconstruction (*La Manno et al., 2018*), in which genes defining B, C, and A cells are expressed sequentially in distinct phases in pseudotime (*Figure 2L–M*).

Overall, our single-cell dataset recapitulates the known B-C-A cell progression through the neurogenic lineage. Intriguingly, our analysis also reveals heterogeneity among B, C, and A cells, in which each cell type is subdivided into multiple distinct clusters. Among C cells, the heterogeneity was mostly driven by different stages of the cell cycle (*Figure 2F–G,I,K*). What drives heterogeneity among B and A cell clusters?

## Quiescent B cell clusters correspond to regionally organized dorsal and ventral domains

In our scRNA-Seq dataset, we found that quiescent B cells (the head of the 'bird'; *Figure 2A*) were subdivided into three clusters: B cell cluster 5 (B(5)), B(14), and B(22) (*Figure 3—figure supplement 1A*). To understand their molecular differences, we conducted differential expression analysis to identify significantly upregulated genes in each of the three B cell clusters (*Figure 3—figure supplement 1B(i)*) and candidate cluster-specific marker genes (*Figure 3—figure supplement 1B(ii)*) (*Supplementary file 3*). When we examined the top ten candidate markers for each cluster, we found genes corresponding to known markers of dorsal and ventral B cell identity (*Figure 3—figure supplement 1C*). For example, *Nkx6.2*, a transcription factor expressed in the ventral embryonic and postnatal ventricular zone (*Merkle et al., 2014*; *Moreno-Bravo et al., 2010*), is enriched in cluster B (14), as are *Notum* and *Lmo1* (*Figure 3A*, *Figure 3—figure supplement 1C*; *Borrett et al., 2020*; *Mizrak et al., 2020*). Similarly, *Gsx2*, a marker of dorsal B cells, is a marker of cluster B(5) (*Figure 3B*, *Figure 3—figure supplement 1C*). Another known dorsal marker, *Emx1*, is significantly upregulated in cluster B(22) (*Figure 3C*, *Supplementary file 3*). When we overlay the expression of

these cluster markers on the neurogenic lineage UMAP plot, we find that their expression is largely restricted to cells within a single B cell cluster, and in the case of *Gsx2*, is retained in C cells and early-stage A cells (*Figure 3A–C*). Activated B cells in cluster B(13) also showed the expression of these regional markers but did not separate into multiple clusters at this resolution (*Figure 2A*). The distinct regional signature that drives clustering for quiescent B cells seems to be overshadowed by activation and proliferation-related genes as these primary progenitors progress into the neurogenic lineage.

To confirm regional organization as a major source of heterogeneity in our scRNA-seq dataset, we turned to a previously generated single-nucleus RNA sequencing (sNucRNA-Seq) dataset we had derived from regionally microdissected V-SVZ. We performed single-nucleus sequencing (sNucRNA-Seq) from microdissected V-SVZ subregions of P35 CD1 mice (n=eight males, nine females). We isolated single nuclei from four microdissected quadrants of the V-SVZ: the anterior-dorsal (AD), posterior-dorsal (PD), anterior-ventral (AV), and posterior-ventral (PV) regions (*Figure 3—figure supplement 2A*; *Mirzadeh et al., 2008*). The four region samples (AD, PD, AV, and PV) were then processed in parallel for sNucRNA-Seq (*Figure 3—figure supplement 2B*). Our sNucRNA-Seq dataset contains 45,820 nucleus profiles. The four region samples underwent quality control steps of filtering out low-quality cells and putative doublets (see Materials and methods). Data from each sample were combined and integrated (Seurat v3 *IntegrateData*) (*Stuart et al., 2019*), then clustered as the scRNA-Seq dataset above. The AD and PD samples had a lower number of cells, but higher sequencing depth compared to the ventral samples (*Figure 3—figure supplement 2I*). Cell identities were annotated based on the detection of previously described cell type markers (*Märtin et al., 2019*; *Zeisel et al., 2018*; *Figure 3—figure supplement 2C–E*).

In the sNucRNA-Seq data, we identified 42 clusters, including those corresponding to cell types within the neurogenic lineage: NSPCs (B cells), mitotic intermediate progenitors (C cells), and neuroblasts (A cells) (*Figure 3—figure supplement 2C–E*). Based on the B cell- or astrocyte-specific markers identified in the scRNA-Seq data above, we also identify a parenchymal astrocyte cluster, as well as ependymal cells, striatal neurons, oligodendroglia, microglia, pericytes and vascular smooth muscle cells, endothelial cells, and leptomeningeal cells (*Figure 3—figure supplement 2D–E*). We also found that all four regions contributed to most clusters (*Figure 3—figure supplement 2F–H*).

To test the hypothesis that B cells are dorso-ventrally organized in the V-SVZ, we took advantage of the region-specific microdissection of the sNucRNA-Seq cells (*Figure 3—figure supplement 2*) and metadata Label Transfer to predict scRNA-Seq B cell region identity (*Stuart et al., 2019*). Each B cell was assigned both a dorsal and ventral 'predicted identity' score based on their similarity to dorsal and ventral nuclei (*Figure 3D*). We then calculated the difference between dorsal and ventral scores for each scRNA-Seq B cell. We found that cells within each cluster were strongly dorsal-scoring (green) or ventral-scoring (magenta), with relatively few cells having similar dorsal and ventral prediction scores (gray) (*Figure 3E*). We found that cluster B(14) scored more highly for ventral identity on average, while clusters B(5) and B(22) scored more highly for dorsal identity (*Figure 3—figure supplement 1D*).

To investigate the potential dorso-ventral spatial organization of dorsal-scoring clusters B(5) and B(22) in vivo, we performed RNAscope in situ hybridization for the differentially expressed transcripts *Small Nucleolar RNA Host Gene 15* (*Snhg15*, a lncRNA), and *Contactin-Associated Protein 2* (*Cntnap2*), which are upregulated in clusters B(5) and B(22), respectively (*Figure 3—figure supplement 1E–L*). Overall, we found that *Snhg15* and *Cntnap2* probes did not exclusively localize in B cells, but were also expressed in subsets of A cells, C cells, and ependymal cells, making it difficult to confirm their B cell expression along the dorsal-ventral axis of the V-SVZ (*Figure 3—figure supplement 1F–H, J–L*). To identify markers more specific to B cells within the V-SVZ, and test the hypothesis that B(5) and B(22) both correspond to dorsally localized B cells, we combined them into a single cluster, B(5+22). Looking at this new cluster's predicted dorsal and ventral identity scores, we found that it had a much higher average dorsal predicted identity score, as well as a lower average ventral predicted identity score than when scored separately, nearly the inverse of cluster B(14)'s prediction scores (*Figure 3F*). This unsupervised, unbiased prediction of region identity at the single-cell level, based on sNucRNA-Seq region-specific microdissection, reinforces our observation that scRNA-Seq B cell clusters have strong gene expression signatures of dorsal or ventral V-SVZ identity.

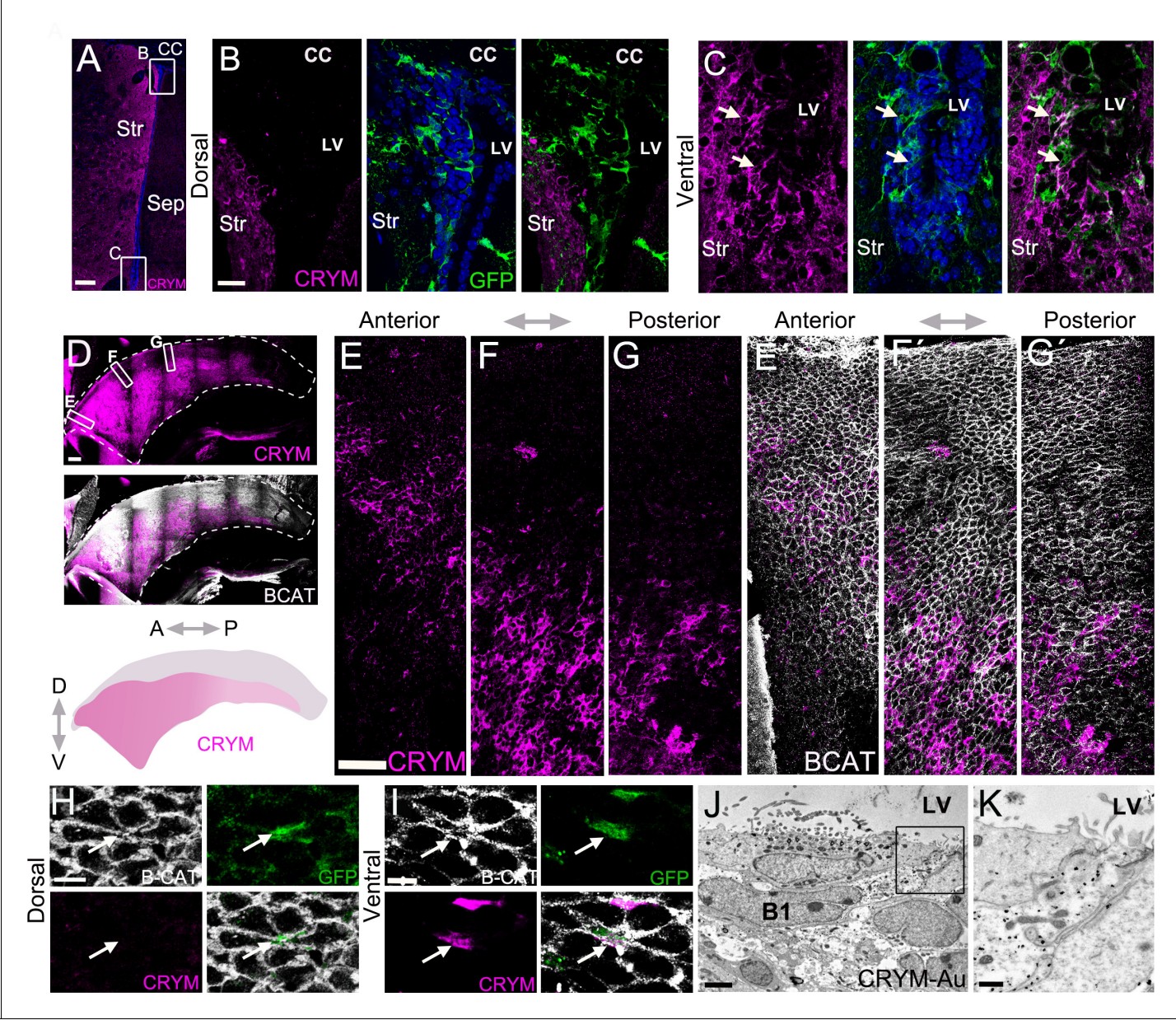

**Figure 4.** Whole mounts reveal CRYM expression in a wide ventral domain. (A-C) Confocal micrograph of a hGFAP:GFP coronal mouse brain section, where the V-SVZ is immunostained for GFP (green) and CRYM (magenta). B. High-magnification image of the dorsal wedge region of the V-SVZ. (C) High-magnification image of the ventral V-SVZ. (D-G) Immunostaining of CRYM (magenta) in a whole-mount preparation of the lateral wall of the V-SVZ, co-stained with β-CATENIN (white), with a summary schematic depicting the extent of the CRYM+ domain. (E–G) Higher magnification images of boxed regions in (D) showing the distribution of CRYM+ (magenta) in V-SVZ cells outlined by β-CATENIN (white). (H-I) High-magnification images of GFP+ (green) B1 cell-containing pinwheels in the dorsal (H) and ventral (I) V-SVZ outlined by β-CATENIN (white), also immunostained for CRYM (magenta). Quantifications showed that 95.11% ± 2.65 (SD) of the GFP+ B1 cells were CRYM+ in the ventral domain of the V-SVZ. In contrast, only 4.71% ± 1.38 (SD) of the GFP+ B1 cells in the dorsal region were CRYM+ (n=3; T-test, p<0.0001). (J-K) Immunogold transmission electron micrographs of CRYM+ B1 cell in the V-SVZ. K. High magnification of B1 cell apical contact with the lateral ventricle. A: anterior, P: posterior, D: dorsal, V: ventral. Scale bars: 150 µm (A), 20 µm (B and C), 200 µm (D), 50 µm (E, F and G), 10 µm (H and I), 2 µm (J), and 500 nm (K).

The online version of this article includes the following source data for figure 4:

**Source data 1.** Quantifications of CRYM+GFP+ B cells in coronal sections and whole mounts.

We then asked what genes were differentially expressed between the putative dorsal cluster B(5 +22) and the putative ventral cluster B(14) (*Figure 3G–I*). Among the differentially expressed genes were other known markers of dorsal B cells, such as *Pax6 and Hopx* (*Figure 3Hi*, *Supplementary file*

3), as well as novel dorsal-domain marker candidates such as *Urah* and *Dio2* (*Figure 3Hii*, I). We found that *Urah* and *Dio2* were highly expressed among GFAP+ cells in the V-SVZ dorsal 'wedge' (the dorso-lateral corner of the V-SVZ enriched in B, C, and A cells), while lower expression was observed in the intermediate and ventral regions (*Figure 3J–S*). Density plots of RNAscope spot number along the ventricular wall showed a higher number of RNAscope spots in the dorsal V-SVZ for *Urah* and *Dio2* (*Figure 3M,R*). This pattern of expression of these dorsal markers was maintained from anterior to posterior sections (*Figure 3M,N,R,S*). To determine if the dorsal *Urah+* and *Dio2+* domains overlap with the dorsal *Hopx+* B cell population (*Zweifel et al., 2018*), we performed RNAscope in situ hybridization for *Hopx*. *Hopx* was expressed in B cells in the dorsal wedge with no expression in the intermediate and ventral V-SVZ regions (*Figure 3—figure supplement 1M–Q*). Interestingly, low and high *Hopx+* cells were observed in the wedge. *Hopx^{high}* cells showed a subcallosal localization forming a band that extended laterally from the septal corner of the V-SVZ, where Hopx+ cells have been previously described (*Zweifel et al., 2018*; *Figure 3—figure supplement 1N–O, Q*). We confirmed that HOPX protein followed the expression pattern of its mRNA by immunostaining (*Figure 3—figure supplement 1R*).

Cluster B(14) marker *Crym* was expressed in intermediate and ventral V-SVZ GFAP+ cells (*Figure 3T–X*), with very little expression in dorsal V-SVZ and wedge regions (*Figure 3T–X*). Consistent with our single-cell data, A cells identified by DCX expression were *Crym*-negative (*Figure 3T–V*), but a subpopulation of cells in the striatum expressed *Crym* (*Figure 3U–V*), which is consistent with previous work (*Chai et al., 2017*; *Mizrak et al., 2019*). To determine if CRYM protein in the V-SVZ was also expressed in a regional pattern, we used antibody labeling to study its expression in B cells using P28 hGFAP:GFP mice. Consistent with the RNAscope analysis, ventral SVZ GFP+ B cells were CRYM+, while dorsal wedge GFP+ cells were negative (*Figure 4A–C*). We defined two domains in our sections (see Materials and methods, *Figure 3—figure supplement 1S*) and quantified the percent of GFP+ B cells that were CRYM+. There was a sharp difference with only 2.47% GFP+ cells being CRYM+ in the dorsal domain, while 97.67% of the ventral GFP+ B cells were CRYM+. Consistent with our observations using RNAscope (*Figure 3U*), whole-mount analysis (*Mirzadeh et al., 2008*), and CRYM antibody labeling showed how the domain of CRYM expression increased in size from the caudal to rostral V-SVZ (*Figure 4D–G*). Using the hGFAP-GFP mice and staining for ß-Catenin to reveal the apical domain of B1 cells and their pinwheel organization (*Mirzadeh et al., 2008*), we confirmed that CRYM was expressed in GFP+ ventral B1 cells, but was largely absent from B1 cells dorsally (*Figure 4H–I*). Quantifications in these whole mounts showed that 95.11% ± 2.65 (SD) of the GFP+ B1 cells were CRYM+ in the ventral domain of the V-SVZ. In contrast, only 4.71% ± 1.38 (SD) of the GFP+ B1 cells in the dorsal region were CRYM+ (n=3; T-test, p<0.0001). To clearly show at a higher resolution and in sections that B1 cells and their apical process has CRYM expression, we used transmission electron microscopy. Immunogold staining showed that B1 cells in the ventral domain express CRYM (*Figure 4*. J-K). Taken together, the above analysis showed that *Crym* transcript and protein expression defined a wide ventral territory that decreased caudally, and was largely absent from the dorsal V-SVZ including the wedge region.

We then conducted GO analysis on differentially expressed genes between dorsal B(5+22) and ventral B(14) cells and found that differentially expressed genes in ventral B cells had an overrepresentation of genes involved in the response to growth hormone (GO:0060416) and retinal ganglion cell axon guidance (GO:0031290), while dorsal B cells were associated with oligodendrocyte differentiation (GO:0048709), forebrain generation of neurons (GO:0021872), and central nervous system neuron axonogenesis (GO:0021955)(*Supplementary file 3*).

To understand the relationships between differentially expressed genes within dorsal and ventral B cell populations, we used gene regulatory network (GRN) analysis. We constructed GRNs based on predicted interactions between the top 10 markers in dorsal B(5+22) cells or ventral B(14) cells and all other genes expressed in each cluster. Dorsal marker genes and their interaction partners formed one large network of 146 genes and three smaller networks with 11–20 genes each (*Figure 3—figure supplement 3A*), while ventral markers formed two large networks (118 and 91 genes) and three smaller ones containing 20–32 genes each (*Figure 3—figure supplement 3B*). The dorsal network was much more highly interconnected than the ventral network (dorsal clustering coefficient = 0.121, ventral clustering coefficient = 0.005). The central dorsal network was enriched for genes associated with regulation of apoptosis (GO:0042981) and central nervous system axonogenesis (GO:0021955), while ventral networks were enriched for genes associated with cell fate commitment

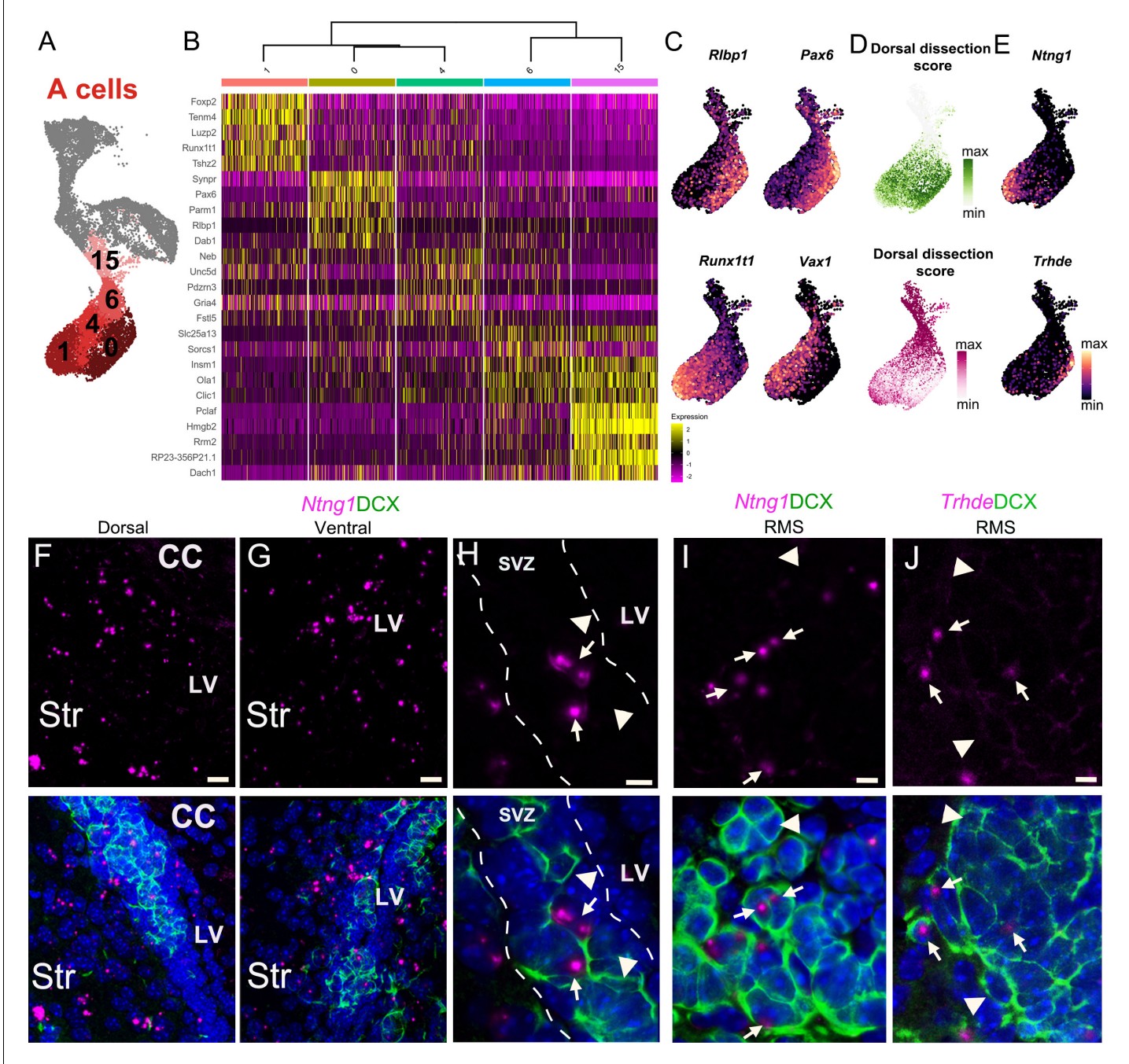

**Figure 5.** A cell cluster heterogeneity is linked to regionally organized dorsal/ventral domains. (**A**) A cells are organized in five distinct clusters (1, 0, 4, 6, and 15). (**B**) Heatmap showing the top five differentially expressed genes for each of the five clusters. (**C**) The expression patterns of *Rlbp1* and *Runx1t1* are similar to *Vax1* and *Pax6*, respectively, which are genes known to be enriched in ventral and dorsal domains of the V-SVZ. (**D**) The predicted regional dissection scores of each scRNA-Seq A cell plotted in UMAP space: cells with strong Dorsal Dissection prediction scores are green and strong Ventral Dissection prediction scores are dark magenta. (**E**) Expression patterns of marker genes *Ntng1* and *Trhde* in A cells. (**F-H**) Expression of *Ntng1* RNA (magenta) and DCX protein (green) in the dorsal (**F**) and ventral (**G**) V-SVZ, and high magnification of *Ntng1*-positive A cells (arrows) and *Ntng1*-negative A cells (arrowheads) in the V-SVZ (dotted lines) (see also *Figure 5—figure supplement 1*). (**I**) High magnification of *Ntng1* puncta in DCX+ A cells (arrows), and *Ntng1*-negative A cells (arrowheads) in the RMS. (**J**) High-magnification image of *Thrde* RNA (magenta) in DCX+ A cells (arrows), along with *Thrde*-negative DCX+ A cells (arrowheads) in the RMS. DAPI: blue, RMS: rostral migratory stream, LV: lateral ventricle, CC: corpus callosum, Str: striatum. Scale bars: 10 μm (**F and G**), 5 μm (**H, I,** and **J**).

The online version of this article includes the following figure supplement(s) for figure 5:

**Figure supplement 1.** Characterization of A cell clusters.

(GO:0045165), regulation of GABAergic synaptic transmission (GO:0032228), and regulation of transcription (GO:0045449). Notably, the dorsal network contained all three genes that make up nuclear receptor subfamily 4A (*Nr4a1*, *Nr4a2*, and *Nr4a3*), members of the steroid-thyroid hormone receptor superfamily that are linked to neuronal specification, axon guidance, and neurotransmission (*Jeanneteau et al., 2018*; *Luo et al., 2008*; *Soldati et al., 2012*). Additionally, phospholipase A2 group VII (*Pla2g7*) is predicted to regulate the ventral marker *Crym* in dorsal cells, likely in a negative direction. To uncover the regulatory context of *Crym* and *Urah*, we built GRNs based on their predicted relationships with genes in putative ventral and dorsal B cells, respectively. We identified two modules related to *Urah* expression in dorsal B(5+22) cells (*Figure 3—figure supplement 4A*) and six related to *Crym* expression in ventral B(14) cells (*Figure 3—figure supplement 4B*). *Urah*-associated modules contained genes related to neurogenesis (GO:0022008), glial cell differentiation (GO:0010001), and hormone metabolic processes (GO:0042445), and *Crym*-associated modules contained genes related to developmental growth (GO:0048589), lipid catabolic processes (GO:0044242), and amine metabolism (GO:0009308). These sets of interacting genes provide clues toward the ways *Urah* and *Crym* may contribute to regional identity. Together, these analyses suggest that differentially expressed genes between these two cell populations are functionally related, working to produce unique cell behaviors that may underlie important differences between B cells in dorsal and ventral domains.

## A cell cluster heterogeneity is linked to regionally organized dorsal and ventral domains

We found that A cells (*Figure 5A*) were separated into two main sets of transcriptionally-related clusters: clusters A(15) and A(6) corresponded to A cells with a strong expression of genes associated with mitosis and cell cycle regulation (such as *Pclaf*, *Hmgb2*, *Rrm2*, *Mki67*, *Top2a*, and the *Mcm* gene family) (*Figure 5B*, *Supplementary file 4*). We calculated the 'area under the curve' (AUC) scores (*Aibar et al., 2017*) for sets of genes corresponding to key GO terms in each A cell. The combined expression of genes in both the GO categories mitotic DNA replication (GO:1902969) and mitosis DNA replication initiation (GO:1902975) were highly upregulated in clusters A(15) and A(6) (*Figure 5—figure supplement 1A*, *Supplementary file 4*). These clusters are also enriched in cells in S and G2M phases (*Figure 2K*), indicating that these clusters correspond to dividing neuroblasts/ early A cells (*Lois and Alvarez-Buylla, 1993*; *Menezes et al., 1995*). Clusters A(1), A(0), and A(4) corresponded to a second set of A cells expressing high levels of genes involved in cell migration, such as *Dab1* and *Slit2* (*Figure 5B*; *Supplementary file 4*). We found that neuron migration (GO:0001764) and spontaneous synaptic transmission (GO: 0098814) categories were more strongly present in clusters A(0), A(1), and A(4) (*Figure 5—figure supplement 1A*), indicating that these clusters likely correspond to migrating young neurons. This analysis of the GO terms enriched in A cells is consistent with the pseudotime analysis (*Figure 2L–M*), indicating that A cells are organized in a continuum of maturation in UMAP space, with dividing neuroblasts at the top of the A cell cluster group and migrating young neurons at the bottom.

Interestingly, the combined expression of genes in dorsoventral axonal guidance (GO:0033563) and cerebral cortex regionalization (GO:0021796), terms associated with the regional specification of the brain, was high in clusters A(1) and A(0), despite individual GO terms not being statistically enriched in these clusters (*Figure 5—figure supplement 1A*; *Supplementary file 4*). Additionally, *Runx1t1*, a transcription factor expressed in young neurons from the medial ganglionic eminence (*Chen et al., 2017*), and *Nxph1*, expressed in young migrating neurons from subpallial germinal zones (*Batista-Brito et al., 2008*), were among the most differentially expressed genes in cluster A(1) (*Figure 5B–C*, *Supplementary file 4*). Conversely, *Pax6*, a transcription factor that is highly expressed in the pallium and in the dorsal lateral ganglionic eminence (*Ypsilanti and Rubenstein, 2016*), was the most differentially expressed gene for cluster A(0) (*Figure 5B–C*, *Supplementary file 4*). Importantly, *Vax1* and *Pax6*, which have been previously found to be differentially expressed by ventrally- and dorsally-born A cells (*Coré et al., 2020*), were highly expressed in clusters A(1) and A(0), respectively (*Figure 5B–C*). Taken together, gene expression suggests that A cells in cluster A(1) originate from ventral progenitors and A cells in cluster A(0) originate from dorsal progenitors. To further support this interpretation, we took advantage of the regional microdissections from the sNucRNA-Seq dataset and scored A cells based on their similarity to A cells from ventral and dorsal microdissections (*Figure 5D*). Among the more mature A cell clusters, we found that cluster A(0)

was enriched in cells that showed high correspondence with A cells from the dorsal dissection, with relatively few cells that had high correspondence with A cells of the ventral dissection (1340 predicted dorsal dissection; 399 predicted ventral dissection). In contrast, cluster A(1) had cells with high correspondence to A cells found in both the ventral and dorsal dissections (925 predicted dorsal dissection; 761 predicted ventral dissection) (*Figure 5D*, *Figure 5—figure supplement 1D*). This is consistent with the inferred patterns of migrations observed in the walls of the lateral ventricles with many A cells from ventral and dorsal origins accumulating anteriorly to join the RMS and a smaller number of cells migrating from dorsal to ventral locations (*Sawamoto et al., 2006*). We do not see a similar mixing of cells in the non-migratory B cell population (*Figure 3E–F*).

In order to confirm that A cells in A(0) and A(1) correspond to dorsal and ventral young neurons, respectively, we looked for markers of A(1) and A(0) that were minimally present in the other cluster. *Trhde* and *Ntng1* were expressed in subpopulations of A cells with high dorsal and ventral dissection scores, respectively (*Figure 5E*). We used RNAscope in situ hybridization with DCX immunolabeling to visualize the localization of transcripts in A cells in vivo. As suggested by the expression pattern in the scRNA-Seq clusters, *Ntng1* was not uniformly expressed in all A cells, but in a subset of them (*Figure 5F–H*), in both dorsal and ventral V-SVZ (*Figure 5F–H*). *Trhde* puncta were present in the V-SVZ, along the ventricular lining, and in areas immediately lateral to the DAPI-dense band of V-SVZ B, C, and A cells (*Figure 5—figure supplement 1B*). This expression pattern is consistent with *Trhde* expression in both the ependymal and striatal neuron scRNA-Seq clusters (*Figure 5—figure supplement 1C*). We found few *Trhde+* DCX+ cells, in either the dorsal or ventral V-SVZ, where each positive A cell had one or two puncta that did not co-localize with the nucleus (*Figure 5—figure supplement 1B*). In addition to *Ntng1 and Trhde* being found in DCX+ A cells dispersed along the full dorso-ventral axis of the V-SVZ (*Figure 5F–H*; *Figure 5—figure supplement 1B*), in the DCX-dense RMS corridor, we found relatively small subsets of A cells that were *Ntng1+* or *Trhde+*, with only one or two mRNA puncta per cell (*Figure 5I–J*). Given the intermixing of A cells as they undergo tangential chain migration in the V-SVZ, the dorso-ventral sites of A cell origins cannot be visualized and validated by immunostaining or RNAscope (*Figure 5F–H*, *Figure 5—figure supplement 1B*). This is consistent with the expected spatial migration patterns of ventrally- vs. dorsally-born A cells (*Fiorelli et al., 2015*; *Sawamoto et al., 2006*). Overall, we found that gene expression patterns, as well as independent, unbiased cell identity prediction provided additional support to the hypothesis that $Pax6^{high}$;$Rlbp1^{high}$ cluster A(0) represents primarily dorsally born A cells, while $Runx1t1^{high}$;$Vax1^{high}$ cluster A(1) represents primarily ventrally born A cells (*Figure 5B–D*). To better understand the differences between clusters A(0) and A(1), we performed a gene ontology (GO) analysis in the differentially expressed genes between these clusters. Differentially expressed genes in A(1) were enriched for genes associated with particular neurite outgrowth and migratory programs, such as regulation of negative chemotaxis (GO:0050923), dorsal/ventral axon guidance (GO:0033563), and V-SVZ-to-OB migration (GO:0022028), suggesting unique and salient features of the migration and guidance preferentially used by ventrally derived A cells. In particular, cluster A(1) was enriched in *Slit1* and *Slit2,* genes that encode guidance cues. Among the differentially expressed genes in cluster A(0), we found an overrepresentation of genes involved in presynaptic membrane assembly (GO:0097105), postsynaptic density assembly (GO:0097107), and synaptic vesicle clustering (GO:0097091) such as *Nrxn1* and *Nlgn1* that were upregulated in dorsally derived A cells. Interestingly, a different set of genes involved in axon guidance (GO:0007411) was upregulated in cluster A(0), including *DCC*, *Efna5*, and *Sema6d.* (*Supplementary file 4*). Together these data suggest that A cells are a molecularly heterogeneous population and that heterogeneity is indicative of an A cell's region of origin within the V-SVZ.

## Specific markers link regionally distinct B cell and A cell lineages

Interestingly, A(1) marker *Slit2* (*Supplementary file 4*) and A(0) marker *Pax6* (*Figure 5B–C*) were also highly expressed in clusters corresponding to ventral and dorsal B cells, respectively (*Figure 3I*; *Supplementary file 3*; see below). In addition, *Rlbp1*, one of the main markers of dorsal B cells (*Figure 3I*), was also strongly expressed in cluster A(0) (*Figure 5C*). To further determine if each dorsal/ventral domain has a specific gene expression signature that persists throughout the neurogenic lineage, we asked which B and A cell subpopulation marker genes were commonly expressed between the ventral B and ventral A cell clusters (B(14) and A(1)), and which were commonly expressed between the dorsal B(5+22) and dorsal A(0) clusters (*Figure 6A–B*). We identified four

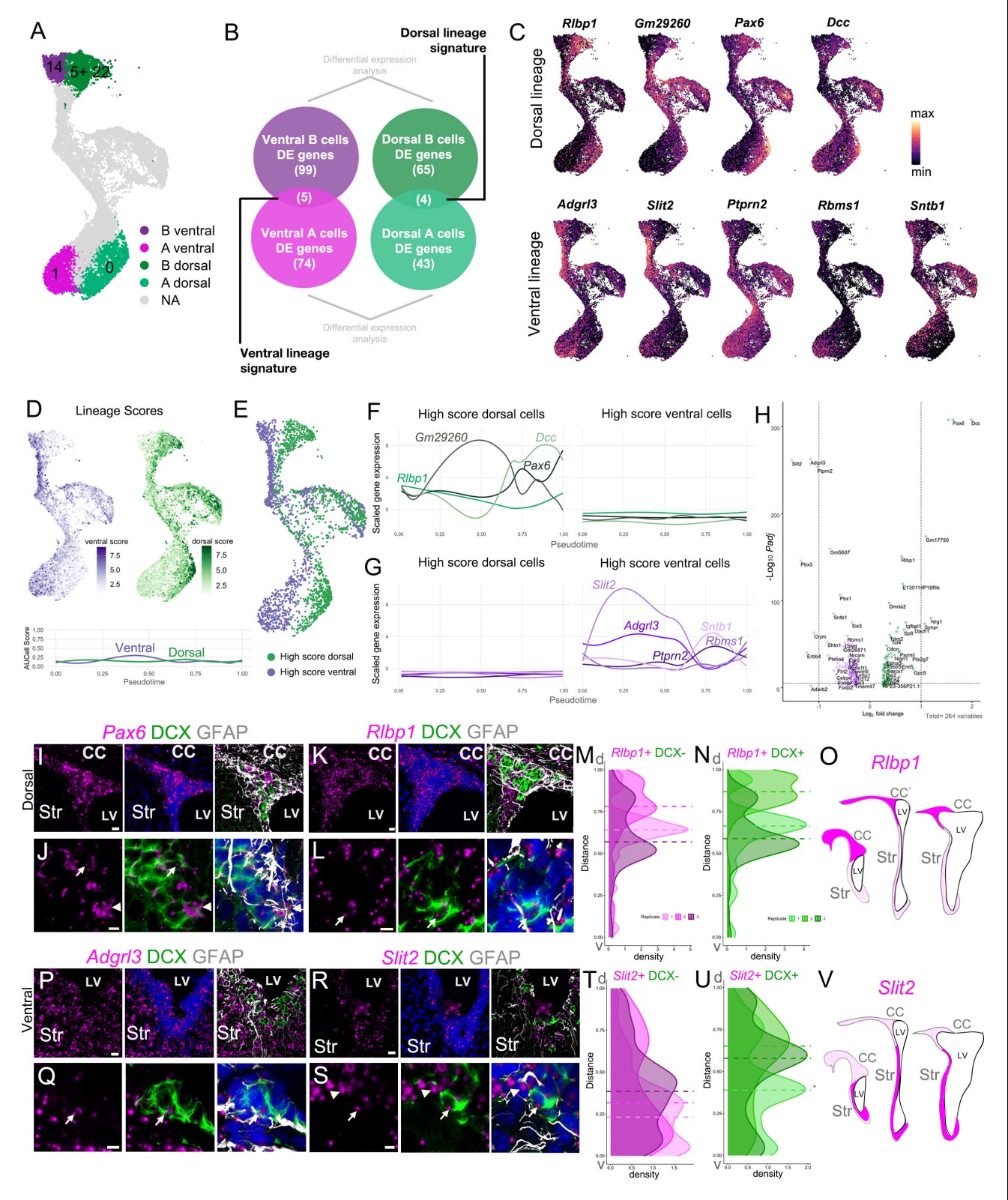

**Figure 6.** Regional transcriptional signatures are maintained along the neurogenic lineage. (**A**) UMAP plot highlighting the putative dorsal and ventral B and A cell clusters. (**B**) Schematic illustrating the approach to identify genes that are differentially enriched in dorsal B and A, and ventral B and A cells, comparing B(14) to A(1) and B(5) and B(22) to A(0). (**C**) Expression patterns of dorsal (top row) and ventral markers (bottom row) identified as differentially enriched throughout the B-C-A lineage. (**D**) Cell scores based on the combined expression of genes in the ventral lineage signature

*Figure 6 continued on next page*

*Figure 6 continued*

(ventral score; purple) and genes in the dorsal lineage signature (dorsal score; green) throughout the neurogenic lineage. Lineage scores, as the combined expression of genes of the dorsal and ventral signatures, remain relatively constant throughout all cells along the neurogenic lineage. (E) High Score (top quartile of each lineage score) dorsal (green) and ventral (purple) cells in the neurogenic lineage. (F) Expression of dorsal signature genes in High Score dorsal (left) and High Score ventral cells (right) along the neurogenic lineage progression, assessed by pseudotime (see also *Figure 6—figure supplement 1*). (G) Expression of ventral signature genes in High Score dorsal (left) and High Score ventral cells (right) along the neurogenic lineage progression, assessed by pseudotime. (H) Volcano plot of significantly differentially expressed genes between High Score dorsal and High Score ventral neurogenic lineages. (I - J) RNAscope validation of dorsal lineage marker *Pax6* (magenta) with DCX (green) and GFAP (white) immunostaining. High-magnification images of the V-SVZ dorsal wedge (I). J. High-magnification image of the V-SVZ where *Pax6* colocalizes with an A cell (arrow) and B cell (arrowhead) (see also *Figure 6—figure supplement 1*). (K- L) RNAscope validation of dorsal lineage marker *Rlbp1* (magenta) in the dorsal wedge (K). L. High-magnification images of *Rlbp1* in the dorsal V-SVZ, where puncta are visible in a DCX-positive A cell (arrow). (M-O) Quantifications of DAPI+,DCX- (M) or DAPI+,DCX+ (N) *Rlbp1* RNAscope puncta along the length of the V-SVZ (0 = ventral-most extent, 1 = dorso-lateral-most extent of the wedge) with the median puncta distribution location plotted as a horizontal line for each sample (n=3, each indicated by a different shade, ~ bregma 1.34, 1.18, and 0.98 mm). (O) Summary schematic of *Rlbp1* expression in three rostro-caudal coronal sections (~bregma 1.50, 0.98, and 0.14 mm); strong expression (magenta); sparse (light magenta). Note that *Pax6* showed a similar distribution. (P-Q) RNAscope validation of ventral lineage marker *Adgrl3* (magenta) in the ventral V-SVZ. (Q) High-magnification image of *Adgrl3* in the ventral V-SVZ, colocalizing with an A cell (arrow). (R-S). RNAscope validation of ventral lineage marker *Slit2* (magenta) in the ventral V-SVZ. S. High-magnification image of the ventral V-SVZ, where *Slit2* puncta colocalize with an A cell (arrow) and B cell (arrowhead). (T-V). Quantifications of DAPI+,DCX- (T) or DAPI+,DCX+ (U) *Slit2* RNAscope puncta along the length of the V-SVZ (0 = ventral-most extent, 1 = dorso-lateral-most extent of the wedge) with the median puncta distribution location plotted as a horizontal line for each sample (n=3, each indicated by a different shade, ~ bregma 1.34, 1.18, and 0.62 mm). V. Summary schematic of *Slit2* expression in three rostro-caudal coronal sections (~bregma 1.50, 0.98, and 0.14 mm); strong expression (magenta); sparse (light magenta). Note that *Adgrl3* showed a similar distribution. CC: corpus callosum, Str: striatum, LV: lateral ventricle. Scale bars: 15 µm (K, P, and R), 10 µm (I), and 5 µm (J, L, Q, and S).

The online version of this article includes the following source data and figure supplement(s) for figure 6:

**Source data 1.** Quantifications of *Rlbp1* and *Slit2* RNAscope spots.
**Figure supplement 1.** Characterization of dorsal and ventral neurogenic lineages.

genes that were expressed throughout the dorsal lineage (*Rlbp1*, *Gm29260*, *Pax6*, and *Dcc*) and five genes expressed through the ventral lineage (*Adgrl3*, *Slit2*, *Ptprn2*, *Rbms1*, *Sntb1*) (*Figure 6C*). The converse comparison, however, of ventral B(14) and dorsal A(0), and dorsal B(5+22) and ventral A(1) clusters, yielded only one or two potential lineage marker genes, respectively (*Figure 6—figure supplement 1A–B*). This suggests that B(14) and A(1), and B(5+22) and A(0) had a higher degree of transcriptional overlap, and correspond to ventral and dorsal lineages, respectively.

To understand the molecular differences between the putative dorsal and ventral lineages, we used the regional gene sets we identified above to calculate a composite AUC score (*Aibar et al., 2017*) for both the dorsal and ventral gene expression signatures (*Figure 6D*, *Figure 6—figure supplement 1C*). We found that cells that scored highly for the dorsal genes were largely located on the right side of the 'bird', and the highest ventral-scoring cells were on its left side (*Figure 6D*, *Figure 6—figure supplement 1C–E*). In order to understand the functional differences between cells in dorsal and ventral lineages, we normalized the dorsal and ventral scores (see Methods) and selected the top-scoring quartile for each lineage (High Score lineages) to compare their gene expression (*Figure 6E*). We found that genes enriched in dorsal or ventral lineages have dynamic expression patterns along pseudotime. For example, *Dcc*, a netrin-1 receptor, has its peak of expression in dorsal A cells; curiously *Slit2*, another guidance molecule, had its peak expression in activated B cells. (*Figure 6C,F,G*, *Figure 6—figure supplement 1F–G*). However, the scores of the combined expression for dorsal and ventral lineage signatures were relatively constant throughout the B-C-A cell lineage progression (*Figure 6D*). We identified 257 significantly differentially expressed genes between the High Score lineages: 108 dorsal markers and 149 ventral markers (*Figure 6H*; *Supplementary file 5*). We then conducted GO analysis on differentially expressed genes between High Score dorsal and ventral cells from clusters B(5+22) and B(14), respectively; and between the High Score dorsal and ventral A cells from clusters A(0) and A(1) (*Figure 6A*, *Figure 6—figure supplement 1*). Among the differentially expressed genes in the High Score dorsal B cells, we found an overrepresentation of the thyroid-stimulating hormone secretion (GO:0070460) with higher expression of *Dio2* and *Slc16a2*. DIO2 converts T4 into the bioactive thyroid hormone T3 and *Slc16a2* encoding MCT8, a major thyroid hormone transporter (*Bernal et al., 2015*). Rostrocaudal neural tube patterning (GO:0021903), and positive regulation of notch signaling pathway (GO:0045747)

categories were also overrepresented. *Hes1* and *Hes5*, commonly used as readouts of Notch activation (*Ohtsuka et al., 1999*), were upregulated in the High Score dorsal B cells. The High Score ventral B cells had an enrichment in genes involved in dorsal/ventral pattern formation (GO:0009953), response to laminar fluid shear stress (GO:0034616), adherens junction assembly (GO:0034333), and cerebrospinal fluid circulation (GO:0090660). We also found an enrichment for signaling pathways in the High Score ventral B cells: PDGFR-signaling pathway (GO:0048008), cellular response to epidermal growth factor stimulus (GO:0071364), and insulin receptor signaling pathway (GO:0008286). Dorsal A cells had a higher expression of *Nlgn1* and *Nrxn1*, genes in the NMDA glutamate receptor clustering category, and also had presynaptic membrane assembly (GO:0097105) and semaphorin-plexin signaling (GO:0071526) categories overrepresented. High Score ventral A cells were strongly enriched in *Slit1*, *Slit2*, and *Robo1,* genes involved in the regulation of negative chemotaxis (GO:0050923). These cells were also enriched in *Cacna1c* and *Slc8a1*, genes involved in calcium ion import across the plasma membrane (GO:0098703) and calcium ion import into the cytosol (GO:1902656) (*Figure 6—figure supplement 1H*; *Supplementary file 5*).

To validate RNA expression of putative dorsal and ventral lineage markers in vivo, we combined RNAscope labeling with GFAP and DCX immunostaining in coronal sections of the V-SVZ. We found that the putative dorsal lineage genes *Pax6* and *Rlbp1* were highly enriched in the dorsal region of the V-SVZ, with particularly enriched expression in the 'wedge' region (*Figure 6I–L,O*; *Figure 6—figure supplement 1I–L*). Density plots of RNAscope spot number along the ventricular wall showed a higher number of RNAscope spots in the dorsal V-SVZ for *Rlbp1*+DCX- and *Rlbp1*+DCX+ cells (*Figure 6M–N*). Conversely, the putative ventral markers *Adgrl3* and *Slit2* had a higher expression in the ventral domain of the V-SVZ (*Figure 6P–S,V*; *Figure 6—figure supplement 1L–O*). Interestingly, density plots of RNAscope spots for *Slit2*+DCX- cells showed higher numbers of spots ventrally, while *Slit2*+DCX+ were equally distributed along the ventricular wall. Consistent with our scRNA-Seq data, these genes were expressed in both B and A cells in the neurogenic lineage (*Figure 6J,L, Q,S*).

## Discussion

During brain development, regional allocations of the neuroepithelium give NSPCs different neurogenic properties. The adult V-SVZ neurogenic niche retains regionally specified NSPCs that generate different subtypes of neurons destined for the OB. A molecular understanding of what makes adult NSPCs different between regions is largely lacking. Our scRNA-Seq and sNucRNA-Seq datasets provide new information about the diverse cell types that populate the V-SVZ. Our lineage analysis reveals parallel pathways of neurogenesis initiated by different populations of B cells. Interestingly, these differences in B cell identity correlate with unique regional patterns of gene expression, which we validated using reference-based metadata label transfer from the second dataset of regionally dissected single V-SVZ nuclei. We confirmed the regional expression of marker genes by immunostaining and RNAscope analysis.

Regional differences in NSPCs potential were demonstrated using restricted viral labeling of non-overlapping territories of the V-SVZ (*Merkle et al., 2014*; *Merkle et al., 2007*; *Ventura and Goldman, 2007*). Labeled ventral B cells produced deep layer granule neurons, calbindin-positive periglomerular cells, and type 1–4 cells in the OB, while dorsal B cells produced superficial granule neurons and tyrosine hydroxylase-positive periglomerular cells (*Merkle et al., 2014*; *Merkle et al., 2007*). Similarly, genetic lineage tracing from territories expressing regionally restricted transcription factors also indicates that NSPCs in dorsal and ventral territories generate superficial and deep-layer neurons for the OB, respectively (*Kohwi et al., 2007*; *Kohwi et al., 2005*; *Merkle et al., 2014*; *Young et al., 2007*). It has been suggested that in the adult V-SVZ, a more primitive population of Oct4+/GFAP- NSCs may be present and that these cells may be earlier in the lineage from the 'definitive' GFAP+ B cells (*Reeve et al., 2017*). However, regionally specified NSPCs can be lineage traced to the embryo (*Fuentealba et al., 2015*; *Furutachi et al., 2015*), and we could not detect a population of Oct4+ cells in our datasets. We, however, cannot exclude that rare primitive OCT4+ NSPCs were not captured in our analysis for technical reasons. More recently, genetic labeling of the most ventral domain of the V-SVZ showed the specific contribution of Nkx2.1-expressing B cells to deep layer granule cell neurons in the OB (*Delgado and Lim, 2015*). The ventral gene expression program is maintained by an epigenetic mechanism across cell divisions. In the absence of myeloid/

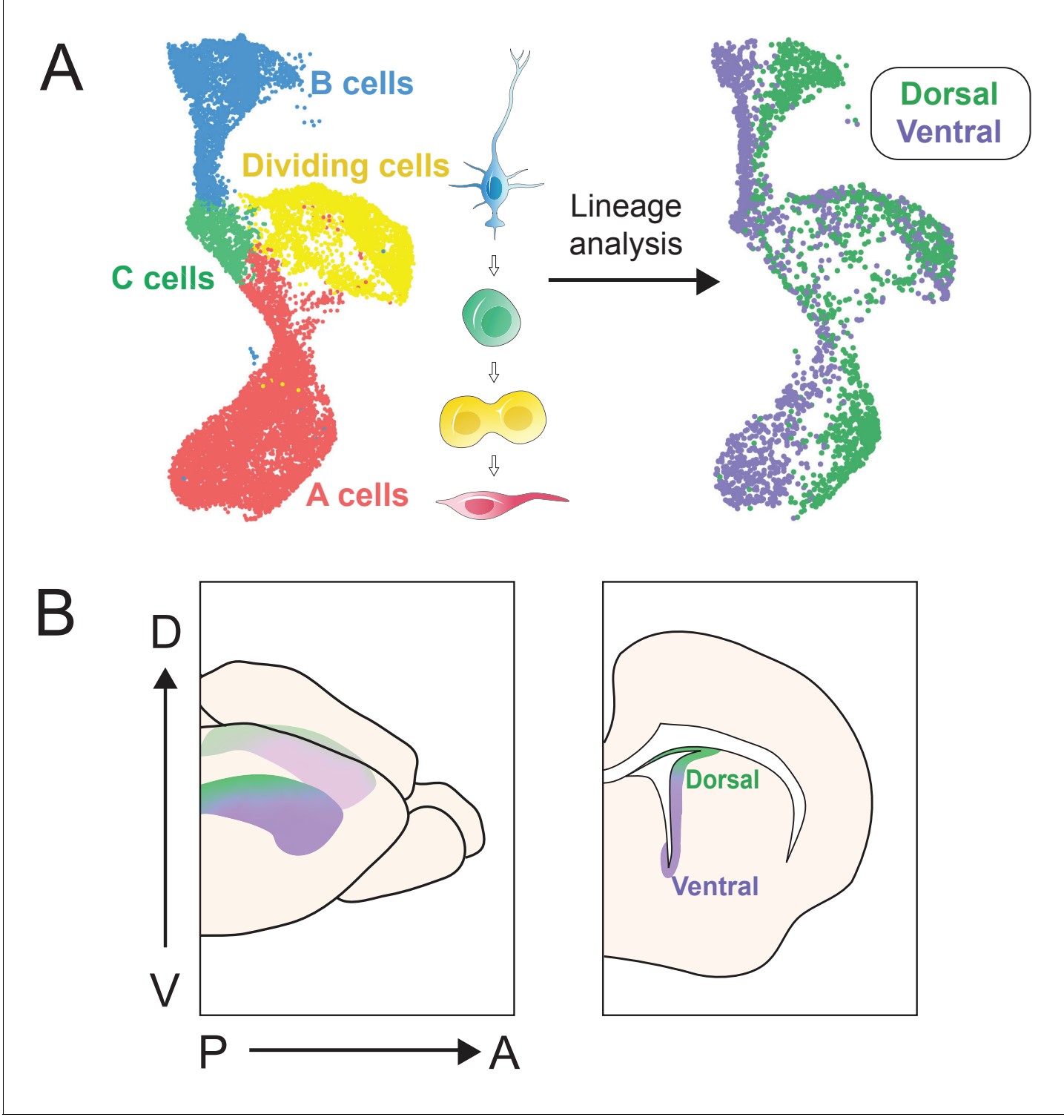

**Figure 7.** scRNA-Seq reveals dorsal and ventral neurogenic lineage domains in the V-SVZ. (**A**) A summary of cell types in the neurogenic lineage identified by scRNA-Seq and their classification into dorsal and ventral transcriptional identities. (**B**) Schematic depicting the dorsal and ventral domains newly identified by scRNA-Seq and snucRNA-Seq, and confirmed by staining and RNAscope.

lymphoid or mixed-lineage leukemia protein 1 (MLL1)-dependent epigenetic maintenance, the neurogenic lineage shifts to produce aberrantly 'dorsalized' OB neuronal subtypes (*Delgado et al., 2020*). This underscores the early embryonic regional specification of adult V-SVZ NSPCs and how

these primary progenitors maintain a memory of their regions of origin. Regional genes maintained through the neurogenic lineage could help us understand how NSC identities are maintained to ensure the production of the specific subtypes of interneurons in the OB.

Our dataset is restricted to the V-SVZ, a region where A cells (young neurons) have begun their differentiation, but remain migratory and immature. As A cells move into the OB they complete their differentiation and begin expressing mature neuronal markers like tyrosine hydroxylase and calbindin in different subdomains of the OB. Our study reveals previously unknown markers of young immature type A cells. Previous work has already shown that Pax6, which is associated with a subpopulation of superficial granule neurons and of periglomerular cells, is expressed by a subpopulation of young migrating A cells (*Coré et al., 2020*; *Kohwi et al., 2005*). Consistent with these findings, we found *Pax6* expression largely restricted in the dorsal lineage of A cells (*Figure 5*). This further validates the regional heterogeneity we find among A cells. The new set of genes associated with young A neurons derived from dorsal or ventral territories should help future studies determine the early programs in the differentiation of specific subtypes of OB neurons. Particularly interesting are sets of chemotaxis genes that are differentially expressed by ventral and dorsal A cells (e.g *Slit2* expression in ventral A cells (*Figure 6*) and *Dab1* in dorsal A cells (*Figure 5*)). These guidance genes may be linked to the different migratory destinies that young neurons need to adopt once they arrive in the OB.

Our dataset provides sets of genes that are differentially expressed in dorsal and ventral B cells. Among these genes, we found well-known regionally expressed transcription factors such as *Pax6*, *Hopx*, *Nkx6.2*, *Gsx2,* and *Vax1* (*Coré et al., 2020*; *Delgado and Lim, 2015*; *Hack et al., 2005*; *Kohwi et al., 2005*; *Merkle et al., 2014*; *Taglialatela et al., 2004*; *Zweifel et al., 2018*). We also identified *Urah, Dio2,* and *Crym* as novel markers that define largely non-overlapping domains of the V-SVZ. *Urah* and *Dio2* define a dorsal domain that includes the wedge and subcallosal roof of the V-SVZ, and *Crym* defines a wide ventral domain. Consistent with the above observations, *Crym* expression has been described in a subpopulation of qNSC in the early postnatal V-SVZ derived from the Nkx2.1 domain (*Borrett et al., 2020*). Our dorsal domain overlaps with that defined by *Pax6* (*Hack et al., 2005*; *Kohwi et al., 2005*) and the ventral domain with that defined by Vax1 (*Coré et al., 2020*). *Pax6* and *Vax1* are transcription factors that link these territories to well-defined embryonic domains involved with the generation of different subsets of neurons in the cortex and striatum. Similarly, *Gsx2* is expressed in a gradient in the embryo, with its highest expression in the dorsal lateral ganglionic eminence (*Corbin et al., 2000*; *López-Juárez et al., 2013*; *Taglialatela et al., 2004*; *Waclaw et al., 2009*; *Young et al., 2007*). Consistent with the dorsal expression pattern, in our dataset *Gsx2* was highly enriched in cluster B(5) (*Figure 3B*). The dorsal region was also enriched in *Ptprz1*, *Hopx*, *Dio2*, *Tnc*, and *Moxd1,* which are also markers of outer radial glia, a subpopulation of human neural stem cells that continue to generate neurons for the cortex after detaching from the pallial wall of the lateral ventricles during prenatal development (*Nowakowski et al., 2016*; *Pollen et al., 2015*). The dorsal V-SVZ domain is likely further subdivided into multiple subdomains. In the current analysis, we pooled together clusters B(5) and B(22) as dorsal. However, the largely pallial marker *Emx1* and dorsal lateral ganglionic eminence marker *Gsx2* were differentially enriched in clusters B(22) and B(5), respectively, suggesting that these two clusters may also represent different sets of regionally specified B cells with distinct embryonic origins. These regions become blurred by cells intermixing in the formation of the wedge region in the postnatal V-SVZ making it difficult to confirm their origin based on expression patterns. In addition to pallial and dorsal subpallial markers, this wedge region likely also includes what has been termed the ventral pallium (*Puelles et al., 2016*), which is characterized in the embryo by the expression of *Dbx1*. Unfortunately, our scRNA-Seq analysis did not detect this marker. Further lineage tracing experiments will help determine the precise embryonic origin and nature of the dorsal V-SVZ, including the wedge region.

We confirmed that B cells defined as dorsal or ventral in our scRNA-Seq were predicted to correspond to dorsal and ventral microdissections using unsupervised label transfer of cell identity from the sNucRNA-Seq data (*Figure 3D–F*). We also found that 59 genes in our scRNA-Seq analysis were highly expressed in sNucRNA-Seq B cells from the dorsal microdissection, including *Pax6*. Similarly, seven genes were highly expressed in the ventral microdissected sNucRNA-Seq B cells that were also highly expressed in the ventral scRNA-Seq B(14) cluster, including *Slit2* (*Supplementary file 6*). However, the identification of single genes in this type of comparison has three limitations: (1) The

dorsal and ventral territories (*Figure 4*; see below) did not precisely correspond to the microdissected areas used for sNucRNA-Seq. For example, regions that we considered dorsal in our microdissection (ventral to the wedge), contained part of the ventral domain highlighted by Crym expression (*Figure 4*). (2) The dorsal samples in the sNucRNA-Seq analysis had fewer cells and higher sequencing depth. (3) mRNA was likely differentially represented in nuclear or whole-cell preps. To overcome these technical issues, we used the unsupervised Label Transfer algorithm to independently identify closely related B cells between the scRNA-Seq and sNucRNA-Seq datasets. We found that cells of cluster B(14) were almost all predicted to correspond to the ventral dissection, and the vast majority of cluster B(5+22) cells were predicted to correspond to cells from the dorsal dissection (*Figure 3E–F*). The above limitations of the label transfer analysis could also affect our observations for A cells (*Figure 5D*).

The border between the *Crym+* and the *Urah/Dio2+* territories had not been previously determined and lies in an anatomically undefined location (*Figure 4*). Interestingly, the dorsal domain defined by *Dio2* and *Urah* largely overlaps with a region of high *Gli1* expression during neonatal and early postnatal stages (*Tong et al., 2015*). This dorsal Sonic Hedgehog-regulated domain in early postnatal life has been linked to oligodendrogenesis. Whether the adult domain we now unravel is developmentally linked to this early oligodendrogenic domain remains to be determined using lineage analysis. However, it is tempting to speculate that these territories are functionally linked, a hypothesis supported by the enrichment of glial-development-associated genes among dorsal markers (*Figure 6G*).

Thyroid hormone signaling has been shown to regulate V-SVZ neurogenesis (*Lemkine et al., 2005*; *López-Juárez et al., 2012*; *Luongo et al., 2021*). Our analysis shows that *Crym, Urah,* and *Dio2* are differentially expressed by B cells according to region. *Crym, Urah,* and *Dio2* are all associated with the thyroid hormone signaling pathway; *Dio2* catalyzes thyroid hormone activation, *Crym* has been described to bind thyroid hormone, and *Urah* belongs to a family of thyroid hormone transporters (*Luongo et al., 2019*; *Rudqvist et al., 2012*; *Vié et al., 1997*). Furthermore, steroid-thyroid hormone family members *Nr4a1, Nr4a2,* and *Nr4a3* were enriched in dorsal B cells and were predicted to form part of the core regulatory network of dorsal marker genes. Hormone signaling, and in particular thyroid hormones, may differentially affect B cells in a regional manner and possibly modify the balance of neuronal types produced in the V-SVZ and destined for the OB. Region-specific regulation of V-SVZ stem cells has been previously suggested for the anterior-ventral V-SVZ, where a subpopulation of proopiomelanocortin hypothalamic projections activates Nkx2.1+ progenitors (*Paul et al., 2017*). The combination of region-specific innervation and molecular composition of B cells could form the basis for a system of hormonal regulation underlying circuit changes in the OB.

In summary, we present a large-scale single-cell description of dorso-ventral identity in the lateral wall of the V-SVZ. Not only do we recapitulate known divisions between dorsal and ventral B cells, but also we identify novel regional B cell markers and uncover gene expression programs that appear to persist throughout lineage transitions (*Figure 7*). These data form a basis for future investigation of NSPCs identity, lineage commitment, and embryonic origin, providing clues to help us understand how molecularly defined stem cell territories are spatially organized, and what distinguishes V-SVZ regions from one another.

## Materials and methods

### Key resources table

| Reagent type (species) or resource | Designation | Source or reference | Identifiers | Additional information |
|---|---|---|---|---|
| Genetic reagent (mouse) | hGFAP:GFP | The Jackson Laboratory (*Zhuo et al., 1997*) | Cat#003257 RRID:IMSR_JAX:003257 | Also referred as hGFAP::GFP: FVB/N-Tg(GFAPGFP)14Mes/J |
| Antibody | Anti-ß-Catenin (Rabbit polyclonal) | Sigma | Cat#C2206 RRID:AB_476831 | IF (1:250) |

*Continued on next page*

Continued

| Reagent type (species) or resource | Designation | Source or reference | Identifiers | Additional information |
|---|---|---|---|---|
| Antibody | Anti-CRYM (Mouse monoclonal) | Santa Cruz | Cat#sc-376687 RRID:AB_11150103 | also referred as anti-u-crystallin IF(1:100) |
| Antibody | Anti-DCX (Rabbit polyclonal) | Cell signalling | Cat#4604S RRID:AB_561007 | IF (1:200) |
| Antibody | Anti-GFAP (Chicken polyclonal) | Abcam | Cat#ab4674 RRID:AB_304558 | IF (1:500) |
| Antibody | Anti-GFP (Chicken polyclonal) | Aves Labs | Cat#GFP1020 RRID:AB_10000240 | IF (1:400) |
| Antibody | Anti-HOPX (Rabbit polyclonal) | Proteintech | Cat#11419–1-AP RRID:AB_10693525 | IF(1:500) |
| Antibody | Anti-S100 (Rabbit polyclonal) | Dako | Cat#Z033 (discontinued) | IF(1:100) |
| Antibody | Anti-Chicken Alexa Fluor 488 (Donkey) | Jackson Immuno Research Labs | Cat#703-545-155 RRID:AB_2340375 | IF(1:500) |
| Antibody | Anti-Chicken Alexa Fluor 647 (Donkey) | Jackson Immuno Research Labs | Cat#703-605-155 RRID:AB_2340379 | IF(1:500) |
| Antibody | Anti-mouse Alexa Fluor 647 (Donkey) | Invitrogen | Cat#A31571 RRID:AB_162542 | IF(1:500) |
| Antibody | Anti-rabbit Alexa Fluor 555 (Donkey) | Invitrogen | Cat#A31572 RRID:AB_162543 | IF(1:500) |
| Antibody | Anti-mouse ultrasmall 0.8 nm IgG | Aurion, EMS | Cat#800.022 RRID:AB_2632382 | IF(1:50) |
| Antibody | Donkey anti-rabbit biotinylated | Jackson Immuno Research | Cat#711-065-152 RRID:AB_2340593 | IF(1:400) |
| Antibody | Anti-CD106 (Rat) oligonucleotide tag | BioLegend | Cat#105725 RRID:AB_2783044 | 1 μg/1 million cells |
| Antibody | Anti-CD24 (Rat) oligonucleotide tag | BioLegend | Cat#101841 RRID:AB_2750380 | 1 μg/1 million cells |
| Sequence-based reagent | MULTIseq barcode | PMID:31209384 | https://gartnerlab.ucsf.edu/more.php | TGAGACCT (A3) |
| Sequence-based reagent | MULTIseq barcode | PMID:31209384 | https://gartnerlab.ucsf.edu/more.php | GCACACGC (A4) |
| Sequence-based reagent | MULTIseq barcode | PMID:31209384 | https://gartnerlab.ucsf.edu/more.php | AGAGAGAG (A5) |
| Sequence-based reagent | MULTIseq barcode | PMID:31209384 | https://gartnerlab.ucsf.edu/more.php | TCACAGCA (A6) |
| Sequence-based reagent | Chromium i7 Multiplex kit | 10x Genomics | PN-120262 | Index numbers B12 and C1 |
| Chemical compound | Red blood cell lysis buffer | BioLegend | Cat#420301 | |
| Chemical compound | Myelin removal beads | Miltenyi Biotec | Cat#130-096-733 | |
| Chemical compound | TSA fluorescein reagent pack | Akoya Biosciences | Cat#SAT701001EA | |
| Chemical compound | TNB Blocking Buffer | Akoya Biosciences | Cat#FP1012 | |
| Commercial assay or kit | Papain Dissociation System-EBSS | Worthington | Cat#LK003150 | |
| Commercial assay or kit | Chromium Single cell 3' Library and Gel Bead Kit v2 | 10x Genomics | Cat#PN-120267 | |
| Commercial assay or kit | Chromium Single cell 3' GEM, Library and Gel Bead Kit v3 | 10x Genomics | Cat#PN-1000092 | |
| Commercial assay or kit | RNAscope 2.5 HD Red Detection kit | Advanced Cell Diagnostics | Cat#320497 | |

Continued

| Reagent type (species) or resource | Designation | Source or reference | Identifiers | Additional information |
|---|---|---|---|---|
| Commercial assay or kit | RNAscope 2.5 HD Duplex Detection kit | Advanced Cell Diagnostics | Cat#322430 | |
| Commercial assay or kit | Mm-Lphn3 (Adgrl3) | Advanced Cell Diagnostics | Cat#317481 | |
| Commercial assay or kit | Mm-Crym | Advanced Cell Diagnostics | Cat#466131 | |
| Commercial assay or kit | Mm-Cnatnp2 | Advanced Cell Diagnostics | Cat#449381 | |
| Commercial assay or kit | DapB | Advanced Cell Diagnostics | Cat#310043 | used as negative control |
| Commercial assay or kit | Mm-Dio2 | Advanced Cell Diagnostics | Cat#479331 | |
| Commercial assay or kit | Mm-Hopx | Advanced Cell Diagnostics | Cat#405161 | |
| Commercial assay or kit | Mm-Ntng1-C2 | Advanced Cell Diagnostics | Cat#488871-C2 | |
| Commercial assay or kit | Mm-Pax6 | Advanced Cell Diagnostics | Cat#412821 | |
| Commercial assay or kit | Mm-PPIB | Advanced Cell Diagnostics | Cat#313911 | used as positive control |
| Commercial assay or kit | Mm-Rlbp1 | Advanced Cell Diagnostics | Cat#468161 | |
| Commercial assay or kit | Mm-Snhg15 | Advanced Cell Diagnostics | Cat#889191 | |
| Commercial assay or kit | Mm-Slit2 | Advanced Cell Diagnostics | Cat#449691 | |
| Commercial assay or kit | Mm-Trhde | Advanced Cell Diagnostics | Cat#450781 | |
| Commercial assay or kit | Mm-Urah-C2 | Advanced Cell Diagnostics | Cat#525331-C2 | |
| Commercial assay or kit | Silver enhancement kit | Aurion, EMS | Cat#25520 | |
| Software, algorithm | Cellranger 2.1.0-v2.3.0 and 3.0.2-v3.2.0 | 10x Genomics | RRID:SCR_017344 | |
| Software, algorithm | Seurat V3 (Rstudio package) | https://satijalab.org/seurat/ | RRID:SCR_007322 | |
| Software, algorithm | scVelo | https://scvelo.readthedocs.io | RRID:SCR_018168 | |
| Software, algorithm | Panther | http://www.pantherdb.org | RRID:SCR_004869 | |
| Software, algorithm | GENIE3 | https://github.com/vahuynh/GENIE3 | RRID:SCR_000217 | |
| Software, algorithm | BiNGO | https://apps.cytoscape.org/apps/bingo | RRID:SCR_005736 | |
| Software, algorithm | Cytoscape | https://cytoscape.org | RRID:SCR_003032 | |
| Software, algorithm | sp (Rstudio package) | https://edzer.github.io/sp/ | RRID:SCR_021328 | |
| Software, algorithm | Imaris v9.6–9.7 | Oxford instruments | RRID:SCR_007370 | |
| Software, algorithm | Adobe Photoshop | Adobe | RRID:SCR_014199 | |
| Software, algorithm | Adobe Illustrator | Adobe | RRID:SCR_010279 | |
| Software, algorithm | AUCell 3.13 | https://bioconductor.org/packages/release/bioc/html/AUCell.html | RRID:SCR_021327 | |

## Mice

Mice were housed on a 12 hr day-night cycle with free access to water and food in a specific pathogen-free facility in social cages (up to five mice/cage) and treated according to the guidelines from the UCSF. Institutional Animal Care and Use Committee (IACUC) and NIH. All mice used in this study were healthy and immuno-competent, and did not undergo previous procedures unrelated to the experiment. CD1-elite mice (Charles River Laboratories) and hGFAP::GFP (FVB/N-Tg(GFAPGFP) 14Mes/J, The Jackson Laboratory (003257)) (*Zhuo et al., 1997*) lines were used. Sample sizes were chosen to generate sufficient numbers of high-quality single cells for RNA sequencing, including variables such as sex, and identifying potential batch effects. Biological and technical replicates for each experiment are described in the relevant subsections below.

## Single whole cell sample preparation and multiplexing

Mice received intraperitoneal administration of 2.5% Avertin followed by decapitation. Brains were extracted and 1 mm slices were obtained with an adult mouse brain slicer (Steel Brain Matrix - Coronal 0.5 mm, Alto). Four samples were processed: sample 1: two males P35; sample 2: two males P35; sample 3: two females P29; and sample 4: two females P29. The lateral ventricle walls were microdissected in L-15 medium on ice and the tissue was transferred to Papain-EBSS (LK003150, Worthington). Tissue was digested for 30 mins at 37°C in a thermomixer at 900 RPM. Mechanical dissociation with a P1000 pipette tip (20 s), then fire-polished pasteur pipette was performed for 5 min. Tissue was digested for 10 more min at 37°C, and dissociated with the pasteur pipette for another 2 min. Cells were centrifuged for 5 min, 300 RCF at room temp, and the pellet was resuspended with DNAase/ovomucoid inhibitor according to manufacturer's protocol (Worthington). Cells were incubated in Red blood cell lysis buffer (420301, Biolegend) 3–4 min at 4°C. For MULTI-seq barcoding, cells were suspended with Anchor:Barcode solution (every sample was labeled with a unique barcode: sample 1 Barcode: TGAGACCT ('A3'); sample two barcode GCACACGC ('A4'); sample three barcode AGAGAGAG ('A5'); and sample four barcode TCACAGCA ('A6')) for 5 min at 4°C. A Co-Anchor solution was added and incubated for 5 min (*McGinnis et al., 2019*). Samples were combined and filtered with a FlowMi 40 µm filter (BAH136800040-50EA, Sigma). To remove myelin, the cell suspension was incubated with Myelin Removal Beads (130-096-733, Miltenyi Biotec) (6 µl/brain) for 15 min at 2–8°C. Cells were washed with 0.5% BSA-PBS and transferred to MACS columns (30-042-401 and QuadroMACS Separator 130-090-976, Miltenyi Biotec). The cell suspension was preincubated with TruStain FcX Plus Antibody (BioLegend, Key resources table) on ice for 10 min, then incubated with oligonucleotide-tagged anti-VCAM1 and anti-CD24 antibodies (BioLegend, Key resources table) on ice for 30 min, then washed twice with 0.5% BSA-PBS by centrifugation (5 min, 4°C, 350 RCF) and filtered with a FlowMi 40 µm filter. The effluent was collected and cell density was counted. Cells were loaded into two wells of a 10x Genomics Chromium Single Cell Controller. We used the 10x Genomics Chromium Single Cell 3' Library and Gel Bead Kit v3 to generate cDNA libraries for sequencing according to manufacturer's protocols. GFP expression of isolated cells was observed under an epifluorescence microscope.

## MULTI-seq barcode library preparation and cell assignment

MULTI-seq and antibody TotalSeq barcode libraries were assembled as previously described (*McGinnis et al., 2019*). Briefly, a MULTI-seq primer is added to the cDNA amplification mix. Afterwards, in the first clean-up step using SPRI beads (0.6x) of the standard 10x library prep workflow, the supernatant is saved, transferred to a new tube and a cleanup step using SPRI (1.6x) is performed to eliminate larger molecules. A library preparation PCR is also performed for the MULTI-seq barcodes. The barcode library is analyzed using a Bioanalyzer High Sensitivity DNA system and then sequenced. The code for demultiplexing samples and detecting doublets can be found at https://github.com/chris-mcginnis-ucsf/MULTI-seq, *McGinnis, 2019*.

## Whole cell sequencing data alignment and processing

We pooled gene expression and barcode cDNA libraries from each 10x Genomics Single Cell Controller well (technical replicates, 'Lane') and sequenced them at the UCSF Center for Advanced Technology on one lane of an Illumina Novaseq 6000 machine. A total of 2,892,555,503 reads were aligned using CellRanger 3.0.2-v3.2.0 (10x Genomics) to a custom version of the mouse reference

genome GRCm38 that included the GFP gene (GFP sequence: *Supplementary file 7*). Reads corresponding to oligonucleotide-tagged TotalSeq antibodies were assigned to cells in CellRanger according to manufacturer instructions.

To identify cell barcodes that most likely corresponded to viable cells, we performed quality control and filtering steps. We excluded cells outside of the following thresholds: UMI count depth: 5th and 95th percentiles; number of genes per cell: below 5th percentile; percentage of mitochondrial gene reads per cell: greater than 10%. We classified cells into sample groups and identified doublets using MULTI-seq barcode abundances (*McGinnis et al., 2019*). We used Seurat Integration (Seurat 3) canonical correlation analysis (CCA) to reduce data dimensionality and align the data from technical replicates (Lane 1 and Lane 2) (*Stuart et al., 2019*).

## Single nucleus sample preparation

Brains were extracted and 0.5 mm slices were obtained. We microdissected the anterior ventral, anterior-dorsal, posterior-ventral and dorsal V-SVZ regions of 17 P35 CD1 male (8) and female (9) mice. Briefly, we used a brain matrix to cut one millimeter thick coronal slabs of the mouse forebrain and used histological landmarks to identify each sampling area (e.g. anterior region landmarks: septum; posterior regions: hippocampus). Regions were dissected under a microscope to reduce the amount of underlying striatum in each sample. Each micro-dissected V-SVZ region was processed in parallel as a distinct sample. We processed tissue samples for nucleus isolation and sNucRNA-Seq as previously described (*Velmeshev et al., 2019*). Briefly, we generated a single nucleus suspension using a tissue douncer (Thomas Scientific, Cat # 3431D76) in nucleus isolation medium (0.32M sucrose, 5 mM CaCl2, 3 mM MgAc2, 0.1 mM EDTA, 10 mM Tris-HCl, 1 mM DTT, 0.1% Triton X-100 in DEPC-treated water). Debris was removed via ultracentrifugation on a sucrose cushion (1.8M sucrose, 3 mM MgAc2, 1 mM DTT, 10 mM Tris-HCl in DEPC-treated water) in a thick-walled ultracentrifuge tube (Beckman Coulter, Cat # 355631) and spun at 107,000 RCF, 4°C for 150 min. The pelleted nuclei were incubated in 250 µL PBS made with DEPC-treated water on ice for 20 min. The resuspended pellet was filtered twice through a 30 µm cell strainer. We counted nuclei with a hemocytometer to determine nucleus density, and loaded approximately 12,000 nuclei from each sample into its own well/lane of a 10x Genomics Chromium Single Cell Controller microfluidics instrument. We used the 10x Genomics Chromium Single Cell 3' Library and Gel Bead Kit v2 to generate cDNA libraries for sequencing according to manufacturers' protocols. We measured cDNA library fragment size and concentration with a Bioanalyzer (Agilent Genomics).

## Single nucleus sequencing data alignment and processing

We pooled the gene expression cDNA libraries from each single nucleus sample and sequenced them on one lane of an Illumina HiSeq 4000 at the UCSF Center for Advanced Technology. The PV sample was further sequenced to increase sequencing depth. A total of 1,340,031,643 reads were aligned using CellRanger 2.1.0–2.3.0 (AV, AD, PD samples); 3.0.2-v3.2.0 (PV sample) (10x Genomics) to a custom mouse reference genome that includes unspliced 'pre-mRNA' (GRCm38), which we expect to be present in cell nuclei (*Velmeshev et al., 2019*).

To identify cell barcodes that most likely corresponded to viable nuclei, we performed quality control and filtering steps. For each region sample, we excluded nuclei outside of the following thresholds: UMI count depth: 5th to 95th percentiles; number of genes per cell: below 5th percentile; fraction of mitochondrial gene reads per cell (<10%). We used Seurat Integration (Seurat 3) canonical correlation analysis (CCA) to reduce data dimensionality and align the data from each region (*Stuart et al., 2019*).

## Single cell and single nucleus sequencing data normalization and dimensionality reduction

We used Seurat 3 (*Stuart et al., 2019*) to analyze both the Whole Cell and Single Nucleus datasets: for each dataset, cells or nuclei from each 10x Chromium Controller Lane (scRNA-Seq: Lanes 1 and 2; sNucRNA-Seq: AD, AV, PD, PV lanes) were integrated using IntegrateData and normalized using regularized negative binomial regression (SCTransform) (*Hafemeister and Satija, 2019*). We calculated 100 principal components (PCs) per dataset, and used 50 (scRNA-Seq) or 100 (sNucRNA-Seq) to calculate cell cluster identities at five distinct resolutions (0.5, 0.8, 1.0, 1.5, and 2.0) and UMAP

coordinates. The cell cluster identities presented in this manuscript correspond to resolution 1.5 (scRNA-Seq metadata column integrated_snn_res.1.5) or 2 (sNucRNA-Seq metadata column integrated_snn_res.2), and were chosen based on visual correspondence with the expression of known neurogenic lineage markers. Sequenced antibody tags in the scRNA-Seq dataset were separately normalized using NormalizeData (method: CLR) and ScaleData, and are included in the Seurat object 'Protein' assay as 'VCAM1-TotalA' and 'CD24-TotalA'.

## Dual fluorescent in situ hybridization-immunofluoresce

Mouse brains (n=3, P30) were serially sectioned using a Leica cryostat (10-µm-thick sections in Superfrost Plus slides). Sections were incubated 10 min with 4% PFA and washed 3x10 min with phosphate-buffered saline (PBS) to remove OCT. Slides were incubated with ACD hydrogen peroxide for 10 min, treated in 1x target retrieval buffer (ACD) for 5 min (at 96–100°C) and rinsed in water and 100% ethanol. Samples were air dried at 60°C during 15 min and kept at room temperature overnight. The day after, samples were treated with Protease Plus for 30 min at 40°C in the RNAscope oven. Hybridization of probes and amplification solutions was performed according to the manufacturer's instructions. Amplification and detection steps were performed using the RNAscope 2.5 HD Red Detection Kit (ACD, 320497) and RNAscope 2.5 HD Duplex Reagent Kit (ACD, 322430). RNAscope probes used: Mm-Lphn3 (also named Adgrl3) (cat.# 317481), Mm-Rlbp1 (cat.# 468161), Mm-Crym (cat.# 466131), Mm-Pax6 (cat.# 412821), Mm-Slit2 (cat.# 449691), Mm-Cnatnp2 (cat.# 449381), Mm-Urah-C2 (cat.# 525331-C2), Mm-Dio2 (cat.# 479331), Mm-Hopx (cat.# 405161), Mm-Ntng1-C2 (cat.# 488871-C2), Mm-Trhde (cat.# 450781). Mm-Snhg15 was custom made (NPR-0009896, cat.# 889191). DapB mRNA probe (cat.# 310043) was used as negative and Mm-PPIB (cat.# 313911) as positive control. RNAscope assay was directly followed by antibody staining for chicken anti-GFAP (Abcam, ab4674,1:500) and rabbit anti-DCX (Cell signaling, 4604S, 1:200) or rabbit anti-S100 (Dako, Z033, 1:100, *discontinued*) (Key Resources table). Samples were blocked with TNB solution (0.1 M Tris–HCl, pH 7.5, 0.15 M NaCl, 0.5% PerkinElmer TSA blocking reagent) 30 min and incubated in primary antibodies overnight. Samples were washed with PBS-Tx0.1% and incubated with secondary antibodies Donkey anti-Chicken Alexa 647 (Jackson ImmunoResearch, 703-605-155, 1:500) and Donkey anti-Rabbit biotinylated (Jackson ImmunoResearch, 711-065-152, 1:400) in TNB buffer for 1.5 hr. Samples were washed and incubated with Streptavidin HRP (1:200 in TNB solution) for 30 min. Washed 3x5 min and incubated with Fluorescein Tyramide 5 min (1:50 in amplification diluent) rinsed and incubated with DAPI 10 min. Sections were mounted with Prolong glass Antifade Mountant (Invitrogen, P36980).

## Immunohistochemistry

Coronal sections (n=four mice, P30) and whole mounts (n=three mice, P28) were incubated with Immunosaver (1:200; EMS, Fort Washington, PA) for 20 min at 60°C, and then 15 min at RT. Tissue was then incubated in blocking solution (10% donkey serum and 0.2% Triton X-100 in 0.1 M PBS) for 1 hr. followed by overnight incubation at 4°C with the primary antibodies: mouse anti-CRYM (Santa Cruz, sc-376687,1:100), rabbit anti-BETA-CATENIN (Sigma, C2206, 1:250), Chicken anti-GFP (Aves labs, GFP1020, 1:400), Rabbit anti-HOPX (Proteintech,11419–1-AP,1:500). On the next day, sections were rinsed and incubated with Alexa Fluor secondary antibodies. Samples were mounted with Aqua-poly/mount (Polysciences Inc, 18606–20).

## Confocal microscopy

Confocal images were acquired using the Leica Sp8 confocal microscope. Samples processed for RNAscope and immunohistochemistry were imaged at ×20 (low magnification) and ×63 (High magnification). For high-magnification RNAscope images, 10–15 optical sections were acquired sequentially using Leica Application Suite X (LAS X) software.

## Quantifications

For quantifications of RNAscope puncta, the V-SVZ of one coronal section hemisphere was tile-scanned per mouse (n=3). Optical sections were taken through the entire thickness of the section at 0.3-micometer intervals using a 63x objective. The tile-scans were imported into Imaris Image Analysis software (v9.7, Bitplane) to detect RNAscope puncta using the Spots tool. Automatic spot

detection was manually adjusted, and spots were categorized according to colocalization with DAPI and S100-Beta or DCX immunolabeling. Spots outside of the lateral wall V-SVZ were manually removed from the dataset, as were spots in z-planes that lacked antibody labeling (e.g. the antibody did not fully penetrate the section). Finally, a line was drawn using the Measurement tool from the ventral-most point in the lateral wall V-SVZ, through the V-SVZ to the dorso-lateral most extent of the wedge (raw data available at *Figure 3* and *Figure 6—source data 1*). These data were exported from Imaris and the density of spots along the length of the V-SVZ was quantified using the R sp package (v.1.4–5).

To determine the proportion of B cells (identified as GFP+ cells) expressing CRYM in coronal sections, tile-scans of the entire V-SVZ (n=4) were acquired. Coronal sections (1.42–0.98 mm anterior to bregma) were utilized. Cells were manually counted using Imaris Image Analysis software (v9.7, Bitplane). The length of the V-SVZ, including the wedge and the lateral wall of the lateral ventricle, was divided in 3. The dorsal domain was defined as the most dorsal third (V-SVZ wedge and the adjacent lateral wall); the ventral domain encompassed the ventral two thirds (*Figure 3—figure supplement 1S*).

For quantifications of B1 cells expressing CRYM, V-SVZ whole mounts from hGFAP:GFP mice (n=3) were immunostained for GFP, ß-Catenin and CRYM. Images of the apical surface of dorsal and ventral regions (8–10 fields/region) of the lateral wall were acquired by confocal microscopy (Leica SP8). B1 cells were identified by their GFP+ small apical endings demarcated by ß-Catenin. B1 cells expressing CRYM were manually counted using Imaris Image Analysis software. Note that within the wedge region, in the most dorsal domain, there is no ventricle and this region is not visible in whole mount preparation; this region was included in the above quantification in sections, but it is not included in the quantifications from the whole mounts. B1 cell whole mount quantifications are expressed as mean ± SD (standard deviation). Student's test (Excel, Microsoft) was used for pairwise comparison between two groups.

## Transmission electron microscopy

For CRYM pre-embedding immunogold staining, mice (n=2) were perfused with 4% paraformaldehyde (PFA)/ 0.5% glutaraldehyde. Brains were cut into 50 µm coronal sections on a vibratome. Floating sections were incubated with 1% Sodium borohydride in phosphate buffer (0.1M) for aldehyde inactivation, cryoprotected with 25% sucrose and permeabilized by freezing and thawing (5x) in methylbutane on dry ice. Sections were blocked with 0.3% BSAc (Aurion) in PB 0.1M for 1 hr at RT and incubated with mouse anti-CRYM (Santa Cruz, sc-376687, 1:100) 72 hr, 4°C. Sections were rinsed and incubated with goat anti-mouse conjugated to colloidal gold (1:50, UltraSmall, Aurion #25120) for 24 hr at 4°C. Silver enhancement and Gold toning were performed as previously described (*Sirerol-Piquer et al., 2012*). Sections were postfixed with 1% osmium-7% glucose in phosphate buffer 0.1M, dehydrated and embedded in Durcupan (Fluka). Ultrathin sections (70 nm) were cut, stained with lead citrate and examined under TEM (Tecnai Spirit G2, FEI).

## Combined gene expression score

We used AUCell to score cells based on the expression of sets of genes (*Aibar et al., 2017*). For creating the dorsal and ventral scores, we used the dorsal and ventral signature genes (*Figure 6* and *Figure 6—figure supplement 1*). To identify the High Score dorsal and ventral cells, we normalized both scores to values between 0 (min) and 1(max). We subtracted the ventral score from the dorsal score and the cells in the top quartile corresponded to High Score dosal cells and on the bottom quartile corresponded to the High Score ventral cells.

## Differential gene expression, GO analysis, and gene regulatory networks

We used Seurat three functions FindMarkers (two groups) or FindAllMarkers (more than two groups) to identify differentially expressed genes among groups of single cells (p_val_adj < 0.05) using a Wilcoxon rank sum test. For single nuclei, Seurat four function FindMarkers was used to identify differentially expressed genes (p_val < 0.05). For detailed parameters see available code (below). Selection of genes from the resulting lists for further analysis are described in the text. Gene Ontology analyses of differentially expressed genes were performed using a binomial test and comparing

differentially expressed genes against the whole mouse genome using Panther v.16 (*Mi et al., 2021*). GO terms with a false discovery rate lower than 5% were considered statistically significant. To construct Gene Regulatory Networks for dorsal and ventral B cells we used gene network inference with ensemble of trees (GENIE3), a tree regression based method that employs Random Forests to rank regulatory links between genes (*Huynh-Thu et al., 2010*). After calculating link lists from expression matrices in putative dorsal or ventral B cells, we used the top 10 differentially enriched genes in each population to build networks. We selected the top 300 links among genes predicted to be regulators or targets of these markers. To identify regulatory connections of *Urah* in dorsal cells or *Crym* in ventral cells, we identified the top 20 predicted regulator or target genes of *Crym* or *Urah*, then expanded our networks to include regulators and targets of those genes. Node tables containing expression data were generated and imported to Cytoscape for visualization. GO analysis was completed in Cytoscape using the BiNGO app. Networks are available.

### Nuclei to whole cell regional label transfer

B cell label transfer: First, we subsetted quiescent B cells from both the scRNA-Seq and sNucRNA-Seq datasets (clusters B(5), B(14), B(22), and sNucRNA-Seq cluster 7). We generated the reference sNucRNA-Seq B cell dataset that consisted of equal numbers of B cells per region, randomly selected from the middle 50% of cells by number of genes identified per cell (25th-75th percentile of SCT_snn_nFeature). This prevented the region with the most nuclei from dominating the prediction scores, and filtering cells by nFeature prior to downsampling resulted in reproducible prediction scoring, likely due to exclusion of low-quality B cells and doublets not rejected in the full dataset quality control steps. Anterior dorsal and posterior dorsal regions were combined to create the Dorsal reference cell set, and the anterior ventral and posterior ventral were combined to create the Ventral reference cell set. Subsetted scRNA-Seq B cells and filtered sNucRNA-Seq B cell sets were individually normalized using SCTransform. We then ran FindTransferAnchors with the following settings: *reference.assay = 'SCT', query.assay = 'SCT', normalization.method = 'SCT', npcs=30, project.query=T, and dims = 1:30*. We then calculated Dorsal and Ventral predicted identity scores for each scRNA-Seq B cell using TransferData (*Stuart et al., 2019*).

A cell label transfer: The same method as above was applied to scRNA-Seq A cell clusters A(0), A(1), A(4), A(6), and A(15), and sNucRNA-Seq clusters 12 and 29.

### RNA velocity

RNA Velocity in the neurogenic lineage was calculated using scvelo (*Bergen et al., 2020*), using 2000 genes per cell. Moments were calculated using 30 PCs and 30 neighbors. Velocity was estimated using the stochastic model. Pseudotime was plotted using the original UMAP coordinates.

### Data and code availability

The RNA sequencing datasets generated for this manuscript are deposited in the following locations: scRNA-Seq and sNucRNA-Seq GEO Data Series: GSE165555.

Processed data (CellRanger output. mtx and. tsv files, and Seurat Object. rds files) are available as supplementary files within the scRNA-Seq (GSE165554) or sNucRNA-Seq (GSE165551) data series or individual sample entries listed within each data series.

Web-based, interactive versions of the scRNA-Seq and sNucRNA-Seq datasets are available from the University of California Santa Cruz Cell Browser: https://svzneurogeniclineage.cells.ucsc.edu.

The code used to analyze the datasets and generate the figures are available at the following location: https://github.com/AlvarezBuyllaLab/SVZSingleCell (copy archived at swh:1:rev:37402d867adce3ff4295f06e8fd6289e1d3ba075), *Cebrian-Silla et al., 2021*.

## Acknowledgements

We thank Christopher McGinnis and Dr. Zev Gartner for the MULTI-seq barcodes and technical advice. We thank Dr. Aparna Bhaduri, Dr. Alex Pollen and Dr. Dmitry Velmeshev for input on single cell experimental design and analysis. This work was supported by grants from the US National Institutes of Health R37 HD032116, R01 NS028478, R01 NS113910, a generous gift from the John G Bowes Research Fund and the UCSF Program for Breakthrough Biomedical Research, partially funded by the Sandler Foundation (to AAB); R01 NS091544, R01 NS112357 and VA 1I01 BX000252

(to DAL); F32 NS103221; K99 NS121273 (to SAR); F31 NS106868 (to DW); and by the Spanish Generalitat Valenciana and European Social Fund (APOSTD2018/A113) (to ACS). AAB is the Heather and Melanie Muss Endowed Chair and Professor of Neurological Surgery at UCSF. AAB is Cofounder and on the Scientific Advisory Board of Neurona Therapeutics.

## Additional information

### Competing interests

Arturo Álvarez-Buylla: Cofounder and on the Scientific Advisory Board of Neurona Therapeutics. The other authors declare that no competing interests exist.

### Funding

| Funder | Grant reference number | Author |
|---|---|---|
| Generalitat Valenciana | APOSTD2018/A113 | Arantxa Cebrian Silla |
| National Institutes of Health | F32 (NS103221) | Stephanie A Redmond |
| National Institutes of Health | K99 (NS121273) | Stephanie A Redmond |
| National Institutes of Health | F31 NS106868 | David Wu |
| National Institutes of Health | R37 HD032116 | Arturo Álvarez-Buylla |
| National Institutes of Health | R01 NS028478 | Arturo Álvarez-Buylla |
| National Institutes of Health | R01 NS113910 | Arturo Álvarez-Buylla |
| National Institutes of Health | R01 NS091544 | Daniel A Lim |
| U.S. Department of Veterans Affairs | 1I01 BX000252 | Daniel A Lim |
| University of California, San Francisco | John G. Bowes Research Fund and the UCSF PBBR partially funded by the Sandler Foundation | Arturo Álvarez-Buylla |
| National Institutes of Health | R01 NS112357 | Daniel A Lim |

The funders had no role in study design, data collection and interpretation, or the decision to submit the work for publication.

### Author contributions

Arantxa Cebrian Silla, Conceptualization, Data curation, Formal analysis, Funding acquisition, Validation, Investigation, Visualization, Methodology, Writing - original draft, Writing - review and editing; Marcos Assis Nascimento, Benjamin Mansky, Conceptualization, Data curation, Formal analysis, Investigation, Visualization, Methodology, Writing - original draft, Writing - review and editing; Stephanie A Redmond, Conceptualization, Data curation, Formal analysis, Funding acquisition, Investigation, Visualization, Methodology, Writing - original draft, Writing - review and editing; David Wu, Data curation, Funding acquisition, Investigation, Visualization, Methodology, Writing - review and editing; Kirsten Obernier, Conceptualization, Funding acquisition, Investigation, Methodology, Writing - review and editing; Ricardo Romero Rodriguez, Susana Gonzalez Granero, Investigation; Jose Manuel García-Verdugo, Resources, Supervision; Daniel A Lim, Supervision, Funding acquisition, Methodology, Writing - review and editing; Arturo Álvarez-Buylla, Conceptualization, Resources, Supervision, Funding acquisition, Methodology, Writing - original draft, Writing - review and editing

### Author ORCIDs

Arantxa Cebrian Silla ![ORCID] https://orcid.org/0000-0003-4623-7655
Marcos Assis Nascimento ![ORCID] https://orcid.org/0000-0002-9830-9801
Stephanie A Redmond ![ORCID] https://orcid.org/0000-0002-5580-6269
Benjamin Mansky ![ORCID] https://orcid.org/0000-0001-8652-0928
David Wu ![ORCID] https://orcid.org/0000-0002-9030-3667

Kirsten Obernier (iD) http://orcid.org/0000-0002-4025-1299
Susana Gonzalez Granero (iD) https://orcid.org/0000-0003-1531-550X
Daniel A Lim (iD) https://orcid.org/0000-0001-7221-3425
Arturo Álvarez-Buylla (iD) https://orcid.org/0000-0003-4426-8925

### Ethics

Animal experimentation: Mice were housed on a 12h day-night cycle with free access to water and food in a specific pathogen-free facility in social cages (up to 5 mice/cage) and treated according to the guidelines from the UCSF. Institutional Animal Care and Use Committee (IACUC) and NIH. All mice used in this study were healthy and immuno-competent, and did not undergo previous procedures unrelated to the experiment. CD1-elite mice (Charles River Laboratories) and hGFAP::GFP (FVB/N-Tg(GFAPGFP)14Mes/J, The Jackson Laboratory (003257)) (Zhuo et al., 1997) lines were used. Sample sizes were chosen to generate sufficient numbers of high-quality single cells for RNA sequencing, including variables such as sex, and identifying potential batch effects. Biological and technical replicates for each experiment are described in the relevant subsections below.

### Decision letter and Author response

Decision letter https://doi.org/10.7554/eLife.67436.sa1
Author response https://doi.org/10.7554/eLife.67436.sa2

## Additional files

### Supplementary files

- Supplementary file 1. B cells vs. astrocytes: DE genes and GO terms.
- Supplementary file 2. Cell cycle phase summary.
- Supplementary file 3. B cell clusters DE genes and GO terms.
- Supplementary file 4. A cell clusters DE genes and GO terms.
- Supplementary file 5. Dorsal and ventral lineages DE genes and GO terms.
- Supplementary file 6. sNucRNA-seq dorsal and ventral B cells differential gene expression analysis.
- Supplementary file 7. GFP gene sequence.
- Transparent reporting form

### Data availability

The RNA sequencing datasets generated for this manuscript are deposited in the following locations: scRNA-Seq and sNucRNA-Seq GEO Data SuperSeries: GSE165555. Processed data (CellRanger output. mtx and. tsv files, and Seurat Object. rds files) are available as supplementary files within the scRNA-Seq (GSE165554) or sNucRNA-Seq (GSE165551) data series or individual sample entries listed within each data series. Web-based, interactive versions of the scRNA-Seq and sNucRNA-Seq datasets are available from the University of California Santa Cruz Cell Browser: https://svzneurogeniclineage.cells.ucsc.edu The code used to analyze the datasets and generate the figures are available at the following location: https://github.com/AlvarezBuyllaLab/SVZSingleCell (copy archived at https://archive.softwareheritage.org/swh:1:rev:37402d867adce3ff4295f06e8fd6289e1d3ba075).

The following datasets were generated:

| Author(s) | Year | Dataset title | Dataset URL | Database and Identifier |
|---|---|---|---|---|
| Cebrian-Silla A, Nascimento MA, Redmond SA, Mansky B, Wu D, Obernier K, Rodriguez R, Gonzalez-Granero | 2021 | Single cell analysis of the ventricular-subventricular zone reveals signatures of dorsal and ventral adult neurogenic lineages. | https://www.ncbi.nlm.nih.gov/geo/query/acc.cgi?acc=GSE165555 | NCBI Gene Expression Omnibus, GSE165555 |

| | | | | |
|---|---|---|---|---|
| S, Garcia-Verdugo JM, Lim DA, Alvarez-Buylla A | | | | |
| Cebrian Silla A, Nascimento MA, Redmond SA, Mansky B, Wu D, Obernier K, Romero Rodriguez R, Gonzalez Granero S, García-Verdugo JM, Lim DA, Álvarez-Buylla A | 2021 | Single Cell and Single Nucleus RNA Sequencing of the Adult V-SVZ | https://cells.ucsc.edu/?ds=svzneurogeniclineage | UCSC Cell Browser, svzneurogeniclineage |
| Cebrian Silla A, Nascimento MA, Redmond SA, Mansky B, Wu D, Obernier K, Romero Rodriguez R, Gonzalez Granero S, García-Verdugo JM, Lim DA, Álvarez-Buylla A | 2021 | Single cell analysis of the ventricular-subventricular zone reveals signatures of dorsal and ventral adult neurogenic lineages. | https://www.ncbi.nlm.nih.gov/bioproject/PRJNA694949 | NCBI BioProject, PRJNA694949 |

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
