## [Decision Letter]

**Acceptance summary:**

The authors use single-cell and single-nucleus RNA-sequencing to reveal the molecular heterogeneity that underlies regional differences in neural stem cells in the adult mouse ventral subventricular zone , producing two separate datasets: one in which the zone was analyzed in toto and one in which subregions were microdissected into anterior-posterior and dorsal-ventral quadrants. The authors defined distinct subtypes of type A and B cells ere stratified into dorsal and ventral identities, leading to identification of a handful of genes that they conclude constitute a conserved molecular signature for dorsal or ventral lineages. The manuscript will be a great resource for the adult neurogenesis field.

**Decision letter after peer review:**

Thank you for submitting your article "Single-cell analysis of the ventricular-subventricular zone reveals signatures of dorsal & ventral adult neurogenesis" for consideration by *eLife*. Your article has been reviewed by 2 peer reviewers, and the evaluation has been overseen by a Reviewing Editor and Marianne Bronner as the Senior Editor. The following individual involved in review of your submission has agreed to reveal their identity: Derek van der Kooy (Reviewer #2).

Essential Revisions:

Reviewers felt that the work represented a solid contribution, but that more rigorous data analysis and interpretation was warranted. The reviewer comments are detailed in their explanation and request for clarification and requests for new data. Most of these requests can be addressed through editing the text, but it is possible that new analysis of existing data will be required, or that additional data will need to be generated, for instance to include micro dissected tissue from the anatomical regions as mentioned by Reviewer 1. Due to the number of concerns raised by reviewers will we plan to return the manuscript to the reviewers to seek their feedback on the suitability of the revision for *eLife*.

*Reviewer #1 (Recommendations for the authors):*

1. The authors could make a more convincing argument for the regional identity of B cells in the scRNA-seq dataset if they showed more data from the sNucRNA-seq dataset. The authors use the regional microdissection sNucRNA-seq dataset to generate dorsal and ventral predicted location scores for cells in the scRNA-seq dataset. However, the authors do not confirm that the at least some of the genes which are differentially expressed between the scRNA-seq dataset (ie. between group 14 and 5+22) are similarly differentially expressed in dorsal and ventral B cells from the sNucRNA-seq dataset. The authors could strengthen their argument for dorsal and ventral identity by verifying their results in the sNucRNA-seq dataset and presenting more data from that dataset.

2. Ideally, the scRNA-seq dataset would have been microdissected for location identity, but I can understand that there may have been logistical issues which prevented that. Instead, the authors use the regional microdissection sNucRNA-seq dataset to classify B cells and A cells in the scRNA-seq dataset, which is the more robust dataset. To make the story more convincing, the authors could rearrange some of the figures and how they tell their story. The authors can begin by presenting their scRNA-seq dataset, and show that they see heterogeneity in the B cell and A cell groups. They predict this molecular heterogeneity may be a reflection of known region difference in these cell types. Then, they decide to do sNucRNA-seq on microdissected SVZ. Because this dataset is less robust as far as genes expressed, they decide to use the known dorsal/ventral location of B cells and A cells in the sNucRNA-seq dataset to determine whether the heterogeneity observed in the scRNA-seq dataset is due to regional heterogeneity. This storyline makes the experiments seem intentional. In its current form, the story comes across as a bit disjointed. As I was reading the text I was waiting to see data from the microdissected experiment and was a bit let down when I realized that experiment was not the main focus of the manuscript.

3. The authors could strengthen their RNAscope and immunohistochemistry validations with simple quantification. For example, what proportion of dorsal and ventral B cells express newly identified putative dorsal/ventral markers? This suggestion applies to Figure 3, 4 and 6.

4. The data supporting the authors' classification of dorsally- and ventrally-derived A cells could be stronger. Authors note that "the dorso-ventral sites of A cell origins cannot be visualized and validated by immunostaining or RNAscope. This is consistent with the expected spatial migration patterns of ventrally- vs. dorsally-born A cells" (p. 16). If this is true, then how were the authors able to use the microdissected sNucRNA-seq dataset to assign a dorsal/ventral score to the A cells? It seems that the same migration problem would prevent the authors from being able to identify ventrally- vs. dorsally-born A cells in their microdissections. Additionally, the authors claim that "we found that cluster A(0) was enriched in highly dorsal-scoring cells with relatively few highly ventral scoring cells, while cluster A(1) was enriched in highly ventral-scoring cells with some highly dorsal-scoring cells" (p. 15). While the data in Figure 5-S1D are convincing for the dorsal assignment of A(0), it appears from the figure that A(1) is also more dorsal than ventral – the opposite of the authors' conclusion. This may be due to difficulty interpreting the size of the circles on the bubble plot, but the visualization in Figure 5D also appears to show that A(1) consists of almost equal parts dorsal- and ventral-scoring cells.

5. The authors generate a rich scRNA-seq dataset of V-SVZ cell types, including those of the neurogenic lineage, and identify molecular heterogeneity in both B cells and A cells. They ascribe the molecular heterogeneity to regional differences in cellular location (dorsal vs. ventral) and provide evidence for these classifications throughout the paper. First, the authors start out with a somewhat biased perspective of the source of the molecular heterogeneity (regional location), rather than taking a more unbiased approach by analyzing the molecular differences between subtypes. Second, the authors missed an opportunity to use the molecular differences between subtypes to come up with new insights into the biological differences between the subtypes. They begin to discuss some of these differences in the discussion, but they could also include some supporting data in the results, like gene ontology analyses or gene regulatory network analyses. These types of analyses would add novel insights, in addition to confirming a previously discovered phenomenon (regional location derived B cell heterogeneity).

---

## [Author Response]

Essential Revisions:Reviewers felt that the work represented a solid contribution, but that more rigorous data analysis and interpretation was warranted. The reviewer comments are detailed in their explanation and request for clarification and requests for new data. Most of these requests can be addressed through editing the text, but it is possible that new analysis of existing data will be required, or that additional data will need to be generated, for instance to include micro dissected tissue from the anatomical regions as mentioned by Reviewer 1. Due to the number of concerns raised by reviewers will we plan to return the manuscript to the reviewers to seek their feedback on the suitability of the revision for eLife.

We agree with editors and reviewers that additional data analysis and quantifications strengthen our analysis. We now include:

-New immunostainings and RNAscope labeling (included in Figure 3, Figure 4 and Figure 6).

-Transmission electron microscopy analysis of CRYM immunogold staining (Figure 4).

-Quantifications for the RNAscope and immunocytochemical analysis (Figure 3, Figure 6).

-Pseudotype analysis to identify how dorsal and ventral genes change during lineage progression (Figure 6 and Figure 6-Supplement 1).

-GO analysis identifying salient features of dorsal and ventral B and A cells (Figure 6-Supplement 1).

-Gene regulatory network analysis of differentially expressed genes in dorsal and ventral B cells (Figure 3-S3 and S4).

We also changed the order of the text to discuss first the single cell data as suggested by reviewer 1 and have clarified the semantic issues raised by reviewer 2. Our study does not include bulk analysis of RNA expression; all the data is using scRNA-Seq and sNucRNA-Seq.

Reviewer #1 (Recommendations for the authors):1. The authors could make a more convincing argument for the regional identity of B cells in the scRNA-seq dataset if they showed more data from the sNucRNA-seq dataset. The authors use the regional microdissection sNucRNA-seq dataset to generate dorsal and ventral predicted location scores for cells in the scRNA-seq dataset. However, the authors do not confirm that the at least some of the genes which are differentially expressed between the scRNA-seq dataset (ie. between group 14 and 5+22) are similarly differentially expressed in dorsal and ventral B cells from the sNucRNA-seq dataset. The authors could strengthen their argument for dorsal and ventral identity by verifying their results in the sNucRNA-seq dataset and presenting more data from that dataset.

We are thankful for this suggestion. We now use the sNucRNA-Seq data for the Label Transfer algorithm to confirm differences in ventral and dorsal B cells revealed by the scRNA-Seq analysis. This simplified the Results section and strengthened our findings defining two broad sets of B cells that we find are spatially organized in dorsal and ventral domains. Following the suggestion of the reviewer, we have confirmed that 59 genes differentially expressed by dorsal cluster B(5+22) in our scRNA-Seq analysis are also significantly differentially expressed by B cells in the dorsal sNucRNA-Seq microdissection. Of differentially expressed ventral cluster B(14) cell genes, 7 are also differentially expressed in the ventral microdissection sNucRNA-Seq B cells (This data is now included in Supplementary Table 6). We highlight in the Discussion section how the Label Transfer approach strengthens the differential regional genes expression, but also spell-out the limitation of our data sets and of directly comparing scRNA-Seq to sNucRNA-Seq (see page 22).

2. Ideally, the scRNA-seq dataset would have been microdissected for location identity, but I can understand that there may have been logistical issues which prevented that. Instead, the authors use the regional microdissection sNucRNA-seq dataset to classify B cells and A cells in the scRNA-seq dataset, which is the more robust dataset. To make the story more convincing, the authors could rearrange some of the figures and how they tell their story. The authors can begin by presenting their scRNA-seq dataset, and show that they see heterogeneity in the B cell and A cell groups. They predict this molecular heterogeneity may be a reflection of known region difference in these cell types. Then, they decide to do sNucRNA-seq on microdissected SVZ. Because this dataset is less robust as far as genes expressed, they decide to use the known dorsal/ventral location of B cells and A cells in the sNucRNA-seq dataset to determine whether the heterogeneity observed in the scRNA-seq dataset is due to regional heterogeneity. This storyline makes the experiments seem intentional. In its current form, the story comes across as a bit disjointed. As I was reading the text I was waiting to see data from the microdissected experiment and was a bit let down when I realized that experiment was not the main focus of the manuscript.

We agree with the reviewer that the reordering of data in the manuscript makes it easier to follow. We have now removed the sNucRNA-seq data from Figure 1 and instead focus the initial analysis on the scRNA-seq data. We present the sNucRNA-seq data as supplementary data (Figure 3—figure supplement 2) when we describe supporting analysis based on the microdissection and the label transfer algorithm.

3. The authors could strengthen their RNAscope and immunohistochemistry validations with simple quantification. For example, what proportion of dorsal and ventral B cells express newly identified putative dorsal/ventral markers? This suggestion applies to Figure 3, 4 and 6.

This is a very helpful suggestion. We now include the following quantifications for regional gene expression:

RNA expression analysis using RNAscope of B cell regional markers: To confirm the regional expression of *Dio2, Urah* and *Crym* in B cells, we performed new dual RNAscope in situ hybridization-immunohistochemistry assays and analyzed their expression in coronal sections following the V-SVZ in the dorso-ventral axis. The identification of B cells in tissue sections is challenging as the most commonly used marker, GFAP, is very fibrillary and largely localized to the processes. We find that processes from neighboring cells can encase other cells, making definitive identification of all V-SVZ GFAP+ cells unreliable. We, therefore, did the above RNAscope assays in combination with S100ß immuno-staining to exclude the expression in neighboring parenchymal striatal astrocytes and ependymal cells. We found that *Crym*+S100- RNAscope puncta were located in the ventral V-SVZ, while *Urah*+S100ß- and *Dio2*+S100ß- puncta were located in the dorsal V-SVZ, with substantial expression in the wedge region. The new data is now presented in the text pages 10-11 and Figure 3M, R and W.

CRYM immunostainings in sections and wholemounts: For CRYM, for which we have an antibody that works well in immunocytochemistry, we also present coronal section and whole-mount analysis in hGFAP:GFP mice revealing the ventral territory of expression for this protein. Coronal sections allowed us to quantify the proportion of B cells expressing CRYM in the dorsal-wedge area compared to the ventral domain. We found that 2.47% of dorsal B cells expressed CRYM, in sharp contrast to the 97.67% of ventral B cells. To identify the proportion of B1 cells expressing CRYM in dorsal and ventral domains we quantified CRYM+/GFP+ and CRYM-/GFP+ cells in whole mounts. We found that 95.11% ± 2.65 (SD) of the GFP+ B1 cells in the ventra domain were CRYM+, but only 4.71% ± 1.38 (SD) of the GFP+ B1 cells in the dorsal region were CRYM+ (n=3). We also noted in the revised manuscript that the most dorsal domain within the wedge region is not visible in wholemount preparation, but quantifications for this region are included from the RNAscope analysis in coronal sections. We also now include high-resolution transmission electron microscopy and immunogold staining for CRYM showing the expression of CRYM in B1 cells in the ventral domain. This data is now included in the text in pages 11-12 and in Figure 4J-K.

RNA expression analysis using RNAscope of regional lineage markers: We include quantifications of *Slit2* and *Rlbp1* in combination with DCX to determine the expression of these lineage ventral and dorsal markers in DCX+ A cells and DCX- V-SVZ cells. We find that the *Rlbp1*-positive A cells are enriched in the dorsal wedge area of the V-SVZ and *Slit2*+DCX+ A cells are located throughout the dorsal and ventral V-SVZ, while the DCX- cells are concentrated in the dorsal wedge (*Rlbp1*) or ventrally (*Slit2*). This new data is now included in the text page 18 and in Figure 6M, N, T and U.

4. The data supporting the authors' classification of dorsally- and ventrally-derived A cells could be stronger. Authors note that "the dorso-ventral sites of A cell origins cannot be visualized and validated by immunostaining or RNAscope. This is consistent with the expected spatial migration patterns of ventrally- vs. dorsally-born A cells" (p. 16). If this is true, then how were the authors able to use the microdissected sNucRNA-seq dataset to assign a dorsal/ventral score to the A cells? It seems that the same migration problem would prevent the authors from being able to identify ventrally- vs. dorsally-born A cells in their microdissections. Additionally, the authors claim that "we found that cluster A(0) was enriched in highly dorsal-scoring cells with relatively few highly ventral scoring cells, while cluster A(1) was enriched in highly ventral-scoring cells with some highly dorsal-scoring cells" (p. 15). While the data in Figure 5-S1D are convincing for the dorsal assignment of A(0), it appears from the figure that A(1) is also more dorsal than ventral – the opposite of the authors' conclusion. This may be due to difficulty interpreting the size of the circles on the bubble plot, but the visualization in Figure 5D also appears to show that A(1) consists of almost equal parts dorsal- and ventral-scoring cells.

We agree that the A cell “Label Transfer Analysis” could have been described more clearly. We have added clarifying language to the text (Pages 14-15), and have clarified that the Label Transfer is a measure of how likely a cell is to be identified in a Dorsal Dissection or a Ventral Dissection. This is distinct from a cell’s dorsal/ventral position of origin, especially for migratory cells such as A cells. The reviewer is also correct that the Dorsal Dissection/Ventral Dissection proportions for cluster A(1) is much more mixed than cluster A(0). We also now provide in the revised Page 15 the specific numbers of each predicted label (Dorsal or Ventral Dissection) for clusters A(0) and A(1) in the text, and clarify that the ventral-to-dorsal migratory pattern of ventrally-born A(1) cells leads us to expect that A(1) cells would be identified in both Dorsal and Ventral Dissections.

5. The authors generate a rich scRNA-seq dataset of V-SVZ cell types, including those of the neurogenic lineage, and identify molecular heterogeneity in both B cells and A cells. They ascribe the molecular heterogeneity to regional differences in cellular location (dorsal vs. ventral) and provide evidence for these classifications throughout the paper. First, the authors start out with a somewhat biased perspective of the source of the molecular heterogeneity (regional location), rather than taking a more unbiased approach by analyzing the molecular differences between subtypes. Second, the authors missed an opportunity to use the molecular differences between subtypes to come up with new insights into the biological differences between the subtypes. They begin to discuss some of these differences in the discussion, but they could also include some supporting data in the results, like gene ontology analyses or gene regulatory network analyses. These types of analyses would add novel insights, in addition to confirming a previously discovered phenomenon (regional location derived B cell heterogeneity).

We did not begin our analysis with a preconceived notion that B cells would segregate based on location. On the contrary, we were interested in deriving markers for apical and non-apical B cells. We observed, as described in the manuscript, molecular differences in quiescent and activated B cells as previously described (Codega et al., 2014; Zywitza et al., 2019; Dulken et al., 2017). However, we were surprised by the strong regional signatures after unbiased B cells clustering and that is what we decided to focus on for the present study. Again, based on unbiased clustering analysis we then found differences in A cells that could be mapped to their origin in different subregions. We agree with the reviewer that there are likely many other interesting biological differences that could emerge from the rich scRNA-seq data. The submitted manuscript includes Gene Ontology (GO) analysis for B (Figure 1—figure supplement 2D, Figure 6—figure supplement 1H, Supplemental table 3), A cells (Fig6—figure supplement 1H, Supplemental table 4) and the lineage (Supplemental table 5) to guide future possible discoveries, but feel that exploring any of this in detail would be beyond the current focus of the study on regional differences.

In order to further provide insights into the biological differences between the subtypes, we also conducted gene regulatory network (GRN) analysis to understand the interactions between markers of dorsal or ventral B cells with other genes in their cellular context. We built networks around the top markers of dorsal or ventral B cell identity (Figure 3—figure supplement 3), as well as novel markers *Crym* and *Urah* (Figure 3—figure supplement 4). These analyses uncovered sets of genes associated with cell survival and axonogenesis in dorsal B cells and sets of genes related to GABAergic transmission and transcriptional regulation in ventral B cells. All three members of the nuclear receptor subfamily 4A (*Nr4a1, Nr4a2*, and *Nr4a3*), a family of steroid-thyroid hormone receptors, were part of the central network in dorsal B cells, helping inform new hypotheses about the differential role of thyroid hormone signaling in B cell regional identity. In all, our GRN analysis provides a rich resource of gene-gene relationships to generate future hypotheses about how regional marker genes function in their cellular context. This data is now included in the text in pages 12-13.